# DIFFERENTIALLY PRIVATE ADAPTIVE OPTIMIZATION WITH DELAYED PRECONDITIONERS

**Tian Li[♠], Manzil Zaheer[♡], Ziyu Liu[♠], Sashank Reddi[◇], Brendan McMahan[◇], Virginia Smith[♠]**
[♠]Carnegie Mellon University, [♡]Google DeepMind, [◇]Google Research
{litian,ziyuliu,smithv}@cs.cmu.edu,
{manzilzaheer,sashank,mcmahan}@google.com

## ABSTRACT

Privacy noise may negate the benefits of using adaptive optimizers in differentially private model training. Prior works typically address this issue by using auxiliary information (e.g., public data) to boost the effectiveness of adaptive optimization. In this work, we explore techniques to estimate and efficiently adapt to gradient geometry in private adaptive optimization *without auxiliary data*. Motivated by the observation that adaptive methods can tolerate stale preconditioners, we propose differentially private adaptive training with delayed preconditioners ($\text{DP}^2$), a simple method that constructs delayed but less noisy preconditioners to better realize the benefits of adaptivity. Theoretically, we provide convergence guarantees for our method for both convex and non-convex problems, and analyze trade-offs between delay and privacy noise reduction. Empirically, we explore $\text{DP}^2$ across several real-world datasets, demonstrating that it can improve convergence speed by as much as $4\times$ relative to non-adaptive baselines and match the performance of state-of-the-art optimization methods that require auxiliary data.

## 1 INTRODUCTION

Adaptive optimizers such as AdaGrad (Duchi et al., 2011; McMahan & Streeter, 2010) and RMSProp (Hinton et al., 2012) are commonly used to improve convergence speed in machine learning training. However, in privacy-sensitive applications, the benefits of adaptivity may degrade as a result of noise added to the preconditioners to guarantee differential privacy (Li et al., 2022). Prior works typically address this issue by using non-sensitive auxiliary data to approximate the underlying structures of private gradients (Asi et al., 2021; Kairouz et al., 2021a; Li et al., 2022). While this can boost performance, assuming access to informative public data may be unrealistic in many privacy-sensitive applications. In this work, we instead ask: **Can we improve privacy/utility trade-offs in private adaptive optimization *without* accessing auxiliary data**?

A key insight we have in addressing this question is that for many machine learning problems, the gradient geometry may not change drastically during successive steps of optimization (e.g., see Figure 1, which plots successive distributions of preconditioner values). This presents an opportunity to estimate the preconditioners used by adaptive optimizers with smaller noise, by averaging across previous iterates. To this end, we propose $\text{DP}^2$, a differentially private adaptive method that uses historical gradients to construct delayed preconditioners with reduced noise. Despite the simplicity of this approach, we find that it can signifi-

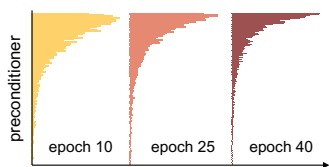

Figure 1: Preconditioner values do not change drastically during optimization (IMDB dataset).

cantly improve performance in practice—improving convergence speed by as much as $4\times$ relative to non-adaptive baselines, all without the need to access auxiliary data. To better understand these performance gains, we theoretically and empirically analyze the method to study the effect of using delayed preconditioners, including trade-offs that emerge between the noise reduction and staleness.

**Contributions.** We propose $\text{DP}^2$ as a method for differentially private adaptive optimization with delayed preconditioners. Unlike prior work, $\text{DP}^2$ does not rely on auxiliary data to improve privacy/utility trade-offs in private training. We provide convergence guarantees for $\text{DP}^2$ in both convex and non-convex settings, and analyze the trade-offs between delay and privacy noise. We conduct

extensive experiments to showcase the effectiveness of $\mathrm{DP}^2$, which can significantly improve model utility for a given privacy budget across text and recommendation benchmarks.

## 2 BACKGROUND AND RELATED WORK

In this section we discuss closely related works and set up some preliminaries. We start by discussing prior work in differentially private optimization, considering the classic framework of $(\varepsilon, \delta)$-differential privacy (DP) (Dwork et al., 2006), defined as follows.

**Definition 1** (Differential privacy (Dwork et al., 2006))**.** *A randomized algorithm $\mathcal{M}$ is $(\varepsilon, \delta)$-differentially private if for all neighboring datasets $D, D'$ differing by one element, and every possible subset of outputs $O$,*

$$\Pr\left(\mathcal{M}(D) \in O\right) \le e^\varepsilon \Pr\left(\mathcal{M}(D') \in O\right) + \delta.$$

**Differentially Private SGD.** Informally, DP in machine learning offers protection by masking the influence of individual examples (example-level DP, e.g. (Abadi et al., 2016; Bassily et al., 2014; Song et al., 2013)) or all of the examples from one user (user-level DP, e.g. (Kairouz et al., 2021b; McMahan et al., 2018)) on the trained model. In this work, we consider example-level DP using the popular subsampled Gaussian mechanism (Dwork et al., 2014; Mironov et al., 2019) to perturb gradients to ensure DP. Unless much larger batch sizes and possibly larger datasets are used, DP mechanisms often lead to a significant utility drop. Extensive research has thus been devoted to investigating improved privacy/utility/computation trade-offs for DP-SGD, including various training techniques (e.g., data augmentation and large-batch training) (De et al., 2022), leveraging public data (Amid et al., 2022; Zhou et al., 2021), and releasing gradient statistics via tree aggregation to reduce the amount of noise (Chan et al., 2011; Denisov et al., 2022; Kairouz et al., 2021b). These prior works are orthogonal to and could be applied in conjunction with our proposed method, which focuses specifically on privacy in the context of adaptive optimization.

**Differentially Private Adaptive Optimization.** To reduce privacy cost in iterative DP algorithms, it is natural to consider applying adaptive optimizers (e.g., AdaGrad (Duchi et al., 2011; McMahan & Streeter, 2010), RMSProp (Hinton et al., 2012), AMSGrad (Reddi et al., 2018), and Yogi (Zaheer et al., 2018)) to speed up convergence. A straightforward approach is to first privatize mini-batch gradients and then plug in noisy gradients to any adaptive updating rules (Zhou et al., 2020). However, estimating gradient moments in this way may yield preconditioners with too much noise, resulting in adaptive methods that may not have meaningful improvements over DP-SGD (Li et al., 2022). As we discuss in Section 1, more recent works suggest the use of non-sensitive public information to estimate the preconditioners (or other gradient structures) (Asi et al., 2021; Kairouz et al., 2021a; Li et al., 2022), which may not always be available in practice. In Section 5.2, we empirically benchmark two baselines along this line of work and demonstrate that $\mathrm{DP}^2$ can perform comparably to these state-of-the-art methods, even though it does *not* require access to auxiliary data. Finally, we note that previous works have explored the high-level direction of delayed preconditioners, but mainly as a compromise for computational considerations in non-private training (Gupta et al., 2018). In this work, we instead show that staleness can be leveraged to improve privacy/utility trade-offs in private adaptive optimization, and propose and analyze a novel method for delaying preconditioner computation in the context of private training.

**Notation.** In this work, we consider using adaptive optimization methods to solve the classic empirical risk minimization objective, i.e., $\min_w F(w) = \frac{1}{n} \sum_{i=1}^n f(x^i; w)$, where $w \in \mathbb{R}^d$ and $\{f(x^i; w)\}_{i \in [n]}$ are individual loss functions on training sample $i \in [n]$. For vectors $u, v \in \mathbb{R}^d$, we use $u + v$ for coordinate-wise addition, and $\frac{u}{v}$ for coordinate-wise division. For any vector $v$, $v_j$ denotes the $j$-th coordinate of $v$. For example, $g_j^{i,t}$ refers to the $j$-th coordinate of gradient $g^{i,t}$. Finally, $|v| \in \mathbb{R}^d$ denotes taking coordinate-wise absolute values, and $\|\cdot\|_M$ denotes the matrix norm defined as $\|\cdot\|_M := \sqrt{\langle \cdot, M \cdot \rangle}$ for a symmetric and positive definite matrix $M \in \mathbb{R}^{d \times d}$, or a diagonal matrix with non-negative diagonal entries populated by a vector $M \in \mathbb{R}^d$.

## 3 $\mathrm{DP}^2$: DELAYED PRECONDITIONERS FOR DIFFERENTIALLY PRIVATE ADAPTIVE OPTIMIZATION

We now introduce our $\mathrm{DP}^2$ framework. While we discuss $\mathrm{DP}^2$ in the context of a particular adaptive method (RMSProp), we note that the approach is method-agnostic in that it can generally be applied

to any private adaptive optimization method where preconditioners are calculated at each iteration. As an initial step towards understanding the algorithm, we first investigate the effects of delayed preconditioners in non-private training in Section 3.1. We then explain how to apply this idea to construct less noisy preconditioners from prior gradients in private training in Section 3.2.

### 3.1 DELAYED PRECONDITIONERS IN NON-PRIVATE SETTINGS

Adaptive methods use preconditioners to adapt to gradient geometry, effectively resulting in coordinate-wise learning rates. This can be advantageous for many applications, especially those with sparse gradients or non-uniform stochastic noise (e.g., Hinton et al., 2012; McMahan & Streeter, 2010; Reddi et al., 2021; Zhang et al., 2020). One of the key design choices of $\mathrm{DP}^2$ is to update preconditioners less frequently and use the average of past gradients to reduce noise. Our observation is that a wide range of learning problems are tolerant to the staleness of preconditioners. In this subsection, we validate this empirically on the benchmark datasets considered throughout this paper.

There are potentially many ways that one could instantiate the idea of delayed preconditioner computation in adaptive optimization. Here we consider a specific algorithm, which is the exact non-private version of our proposed $\mathrm{DP}^2$ framework (Algorithm 1) introduced in later sections. The basic idea is to alternate between $s$ steps of SGD and $s$ steps of an adaptive method (for simplicity we assume RMSProp as the adaptive algorithm), where $s$ is a constant larger than 1. Each time we switch from SGD to RMSProp, we average $s$ past SGD gradients and use the average to update the preconditioner. The preconditioner will be used in subsequent RMSProp updates (thus being stale). As motivation for $\mathrm{DP}^2$, we empirically show that RMSProp with delayed preconditioners achieves almost the same optimization performance as RMSProp (Figure 2).

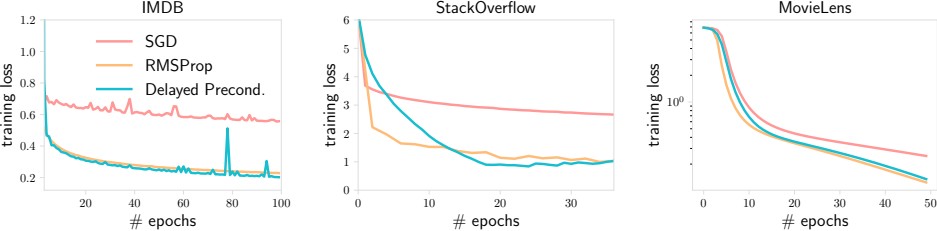

Figure 2: In non-private training, RMSProp with delayed preconditioners achieves similar training loss as standard RMSProp across all datasets. Final test accuracies are presented in Section 5.1. This observation provides motivation for our proposed $\mathrm{DP}^2$ framework for private training (Section 3.2).

As discussed in Section 2, we note that the idea of delayed preconditioning has been briefly discussed in prior work (Gupta et al., 2018) for the purpose of speeding up the computation of adaptive optimization in non-private training. Unlike this prior work, we focus on the goal of reducing noise in private training, propose an alternative method for using stale preconditioners that is more amenable to differential privacy, and analyze our method in both convex and non-convex settings.

### 3.2 CONSTRUCTING DELAYED PRECONDITIONERS WITH REDUCED NOISE

Without access to public data or other side information, prior works typically update preconditioners based on noisy gradients at each iteration (Zhou et al., 2020). For instance, a natural way to privatize RMSProp is to update the preconditioner $v \in \mathbb{R}^d$ as $v \leftarrow \beta v + (1-\beta)(\tilde{g})^2$ where $\beta \in (0,1)$ is a moving average constant, and $\tilde{g} \in \mathbb{R}^d$ is the noisy gradient output by some standard privacy mechanism (e.g., the Gaussian mechanism).[1] However, a drawback to this is that the noise gets accumulated at each iteration, making adaptive methods significantly less effective (Li et al., 2022).

Inspired by the observation that problems can be tolerant to the staleness of preconditioners (Figure 2), we propose to update the preconditioners less frequently to reduce noise. For instance, we update $v$ every $s$ steps using some aggregate function of $s$ recent private gradients from DP-SGD. During iterations where $v$ is not updated, we simply apply the most recent (stale) $v$ to precondition the gradients. In order to mitigate the noise, we *average* over these $s$ gradients to form a pseudo-gradient $g$, which can be plugged into arbitrary adaptive optimization algorithms. Note that the privacy noise variance will be reduced $s$ times if we average $s$ Gaussian random variables (i.e., the DP noise).

---

[1]We consider the practical diagonal (as opposed to matrix) form of adaptive methods throughout the paper.

---

**Algorithm 1:** $\texttt{DP}^2$-RMSprop: Delayed Preconditioners for Differentially Private RMSprop

---

**Input:** $T$, batch size $b$, noise multiplier $\sigma$, clipping thresholds $C$, initial model $w^0 \in \mathbb{R}^d$, $v = \mathbf{0}$,
constant $\epsilon \in \mathbb{R}_+$, learning rate schedule $\alpha^t$, moving average parameter $\beta$, SGD
cumulative aggregation step $s_1$, RMSProp cumulative step $s_2$

---

1 **for** $t = 0, \cdots, T - 1$ **do**
2     **if** $t \bmod (s_1 + s_2) = 0$ **then**
3        Reset accumulator $G^t \leftarrow \mathbf{0}$
4     **if** $t \bmod (s_1 + s_2) = s_1$ **then**
5        Update moment estimates as $v \leftarrow \beta v + (1 - \beta) \left(G^t / s_1\right)^2$
6        Reset accumulator $G^t \leftarrow \mathbf{0}$
7     Uniformly randomly sample a mini-batch $B$ with size $b$ from private training data
8     Get individual gradients for sample $i \in B$: $g^{i,t} \leftarrow \nabla f(x^i; w^t)$
9     Privatize the (preconditioned) gradients using the Gaussian mechanism:

$$\tilde{g}^t \leftarrow \frac{1}{b} \left( \sum_{i \in B} \text{clip} \left( \frac{g^{i,t}}{D^t}, C \right) + \mathcal{N} \left( \mathbf{0}, \sigma^2 C^2 \right) \right)$$

      where
$$D^t \leftarrow \begin{cases} \mathbf{1} & \text{if } t \bmod (s_1 + s_2) < s_1 \\ \sqrt{v} + \epsilon & \text{otherwise.} \end{cases}$$

10     Accumulate the private gradients $\tilde{g}^t$: $G^{t+1} \leftarrow G^t + \tilde{g}^t$
11     Update model parameters $w$:
$$w^{t+1} \leftarrow w^t - \alpha^t \tilde{g}^t$$

12 **return** $w^T$

---

$\texttt{DP}^2$ is summarized in Algorithm 1. For simplicity of presentation, we assume RMSProp as the adaptive method (denoted as $\texttt{DP}^2$-RMSProp) throughout this section. However, our framework can be generally applied to other common adaptive methods (see Appendices C.3 and D). The high-level idea is to alternate between $s_1$ steps of private SGD and $s_2$ private RMSProp steps, and use averages of $s_1$ SGD gradients (i.e., average of the accumulator $G \in \mathbb{R}^d$) to update the preconditioner $v$. Next, we discuss some key components of our algorithm.

**Order of privatization and preconditioning.** Given a private preconditioner $v$, there are generally two choices to perform adaptive optimization over the raw gradients $\{g^{i,t}\}_{i \in B}$ generated from mini-batch $B$ at the $t$-th iteration.

1. First privatize gradients with clipping threshold $C_1$, then precondition noisy gradients with $\sqrt{v} + \epsilon$ where $\epsilon$ is a small constant:

$$\tilde{g}^t \leftarrow \frac{1}{b} \left( \sum_{i \in B} \text{clip} \left( g^{i,t}, C_1 \right) + \mathcal{N} \left( \mathbf{0}, \sigma^2 C_1^2 \right) \right) / \left( \sqrt{v} + \epsilon \right)$$

2. First precondition gradients with $\sqrt{v} + \epsilon$, then privatize the output with clipping threshold $C_2$:

$$\tilde{g}^t \leftarrow \frac{1}{b} \left( \sum_{i \in B} \text{clip} \left( g^{i,t} / \left( \sqrt{v} + \epsilon \right), C_2 \right) + \mathcal{N} \left( \mathbf{0}, \sigma^2 C_2^2 \right) \right)$$

The difference is that the privacy noise in the first choice may be scaled in an undesired direction, as $\frac{\mathcal{N}(\mathbf{0}, \sigma^2 C^2)}{\sqrt{v} + \epsilon}$ with a *less noisy* estimated $\sqrt{v}$ (perfect estimation removing all privacy noise in the extreme case) would amplify the noise $\mathcal{N}(\mathbf{0}, \sigma^2 C^2)$ on informative coordinates (i.e., coordinates with smaller preconditioner values), which is consistent with the argument made in Li et al. (2022). We empirically compare the two options and show that the latter gives better performance (Section 5.3).

It is critical to *average* noisy gradients to construct a cleaner estimate of the preconditioner (Line 5 and 10 in Algorithm 1) and apply it for adaptive optimization (Line 9). As these two steps access raw gradients twice, we need to privatize them separately. Unfortunately, the privacy budget would accumulate with each query to the raw training data. Hence, we use the private SGD gradients for both

the model update and the preconditioner estimation. This results in a hybrid method that alternates between private SGD and private adaptive optimization steps. Note that to get an unbiased estimate of the true delayed preconditioners, we can correct the bias in $(G^t/s_1)^2$ (Line 5) by subtracting the privacy noise variance term $\frac{\sigma^2 C^2}{s_1 b^2}$ out of $(G^t/s_1)^2$. But this value is usually very small and negligible in practice. While in principle, non-adaptive and adaptive updates can take different numbers of consecutive iterations, in our empirical evaluation, we simply set $s_1 = s_2$, and find that this works reasonably well across all datasets (Section 5).

**Privacy guarantees.** From Algorithm 1, we see that at each iteration, we access raw data and pass them through the privacy barrier *once* (Line 9) to generate private gradients $\tilde{g}^t$ with the same noise multiplier $\sigma$ and batch size $b$, and the preconditioner only accumulates already differentially private gradients. Since the final model is a composition of these private releases (noisy gradients), Algorithm 1 (or $\mathrm{DP}^2$ in general) achieves the same privacy guarantees as standard DP-SGD training under the same training settings. For completeness, we formally state the privacy guarantee below.

**Theorem 1** (Privacy guarantee of Algorithm 1 (Abadi et al., 2016)). *There exist constants $c_1$ and $c_2$ such that for any $\varepsilon < c_1 b^2 T / n^2$, Algorithm 1 is $(\varepsilon, \delta)$-differentially private for any $\delta > 0$ if $\sigma \geq c_2 \frac{b\sqrt{T \log(1/\delta)}}{n\varepsilon}$.*

In practice, we use Rényi differential privacy (RDP) for the subsampled Gaussian mechanism accountant (Mironov et al., 2019) to compute the actual $\varepsilon$'s reported in the experiments (Section 5).

## 4 CONVERGENCE ANALYSIS

In this section, we analyze Algorithm 1 for both convex and non-convex problems. We aim to study the convergence properties of $\mathrm{DP}^2$ and investigate the trade-offs between delay and privacy noise. In doing so, key challenges are introduced by alternating between adaptive and non-adaptive updating and through the staleness of preconditioners.

### 4.1 CONVEX CASES

For convex functions, we define the optimal model $w^*$ as $w^* \in \arg\min_w F(w)$. First we state some assumptions (apart from convexity) that are used in the analysis.

**Assumption 1.** *There exists a constant $R$ such that $\|w^t - w^*\|_2 \leq R$ for any iteration t.*

**Assumption 2** (Bounded stochastic gradient norm). *There exists a constant $C$ such that $\left\|g^{i,t}\right\|_2 \leq C$ for any $i \in [n]$ and iteration t.*

Assumption 1 (bounded domain across all iterations) is commonly used in adaptive optimization literature (Asi et al., 2021; Levy et al., 2018; Li et al., 2022; Reddi et al., 2018). Assumption 2 aims to bound the $L_2$ norm of the stochastic gradient, thus helping bound the $L_2$ sensitivity of the operation of calculating and averaging individual gradients from a mini-batch. Assuming bounded stochastic gradient norm is standard in prior works on convex and non-convex private optimization (e.g., Kairouz et al., 2021a; Li et al., 2022; Zhou et al., 2020). Under this assumption, suppose the clipping does not happen, we have $\tilde{g}^t \leftarrow g^t + \mathcal{N}(0, \sigma^2 C^2/b^2)$, where $g^t := \frac{1}{b} \sum_{i \in B} g^{i,t}$. Without loss of generality, let $s_1 = s_2$ in Algorithm 1. Our main convergence result is as follows (assuming $t$ starts from 1).

**Theorem 2** (Convergence of Algorithm 1 for convex problems). *Let Assumptions 1 and 2 hold. Assume $F$ is a convex function. Let the learning rate $\alpha^t$ be set as $\alpha^t \leftarrow \frac{\alpha^{\left\lfloor \frac{t}{2s} \right\rfloor + \left\lfloor \frac{t+s}{2s} \right\rfloor + 1}}{\sqrt{t}}$. After running Algorithm 1 for $T$ iterations with $s = \upsilon T$ for a small constant $\upsilon \in (0, 1]$, we obtain*

$$\min_{t \in [T]} \mathbb{E}\big[F(w^t)\big] - F(w^*) \leq \frac{R^2 + \kappa}{\alpha^{\left\lfloor \frac{1}{2\upsilon} \right\rfloor + \left\lfloor \frac{1+\upsilon}{2\upsilon} \right\rfloor}} \frac{1}{\sqrt{T}} \sum_{t \in T_\upsilon} \mathbb{E}\big[\big\|D^t\big\|_1\big] + \frac{1}{T} \sum_{t=1}^{T} \frac{\alpha^{\left\lfloor \frac{t}{2\upsilon T} \right\rfloor + \left\lfloor \frac{t+\upsilon T}{2\upsilon T} \right\rfloor}}{\sqrt{t}} \mathbb{E}[\|N^t\|_{D^t}^2],$$

*where $T_\upsilon$ denotes the iteration indices where we switch from private RMSProp steps to private SGD steps plus the last iteration, with cardinality $|T_\upsilon| = \lceil \frac{1}{2\upsilon} \rceil$, and $N^t \sim \mathcal{N}(\mathbf{0}, \sigma^2 C^2/b^2)$, and*

$$\kappa \geq \max\left\{\alpha^2 C^2, \frac{Ch(s)}{\epsilon\sqrt{1-\beta}}\right\}, \quad \alpha = \min\left\{\epsilon, \frac{1}{\sqrt{M}+\epsilon}, 1\right\} \quad \text{where } M := C^2 + \frac{\sigma^2 C^2}{sb^2}.$$

We defer all proofs to Appendix A and state simplified convergence results in Corollary 1. As we can see, the above upper bound relies on a critical metric $h(s)$ which is related to temporal gradient similarity and the amount of staleness $s$, formally defined as:

$$h(s) \geq \max_{t \in [T]} \frac{\mathbb{E}\left[\|g^t\|_1\right]}{\mathbb{E}\left[\left\|\frac{1}{s} G^{\lfloor \frac{t}{s} \rfloor s}\right\|_1\right] + d\epsilon} = \max_{t \in [T]} \frac{\mathbb{E}\left[\|g^t\|_1\right]}{\mathbb{E}\left[\frac{1}{s}\left\|\sum_{i=\lfloor \frac{t}{s} \rfloor s - s}^{\lfloor \frac{t}{s} \rfloor s - 1} \tilde{g}^i\right\|_1\right] + d\epsilon},$$

where the expectation is taken with respect to all randomness in the algorithm, and $G^{\lfloor \frac{t}{s} \rfloor s} \in \mathbb{R}^d$ refers to the latest accumulator that is used to update $v$ (Line 5 in Algorithm 1). A smaller $h(s)$ indicates better convergence. We see that the denominator of $h(s)$ can be decomposed into the average of past raw gradients and the average of random Gaussian noise. Intuitively, $h(s)$ tends to be smaller as gradients across the $s$ iterations in $G^{\lfloor \frac{t}{s} \rfloor s}$ are more similar with the current gradient $g^t$ in terms of the gradient norms. In Appendix A.2, we show that an upper bound of $h(s)$ can be expressed as $c_1 + c_2 s$ where $c_1, c_2$ are two constants. We also visualize the value of $h(s)$ on the IMDB dataset in Figure 3, and show that (1) the values of $h(s)$ are consistently small across all delays, and (2) $h(s)$ increases as the $s$ gets larger, which is consistent with the expression of $s$.

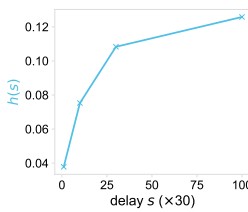

Figure 3: Visualization of $h(s)$ versus $s$ on IMDB.

**Trade-offs between delay and noise.** Here we discuss how $s$ affects convergence based on our analysis. Intuitively, larger $s$ (larger delay) results in staler preconditioners, but introduces less noise due to private gradient averaging. In our convergence bound, there are several terms that depend on $s$ (or $v$). Although this makes it difficult to derive a closed-form characterization of an optimal $s$, we can analyze the effects of $s$ in simplified settings. In particular, examine the first term of the RHS of the convergence bound, let $\alpha = \frac{1}{\sqrt{M}+\epsilon} = \frac{1}{\sqrt{c_3 + \frac{c_4}{v}} + \epsilon}$ (where $c_3, c_4$ are two constants), and assume $\lfloor \frac{1}{2v} \rfloor + \lfloor \frac{1+v}{2v} \rfloor = \frac{1}{2v} + \frac{1+v}{2v} = \frac{2+v}{2v}$. Combined with $h(s)$, the dependence on $v$ in $\frac{R^2+\kappa}{\alpha^{\lfloor \frac{1}{2v} \rfloor + \lfloor \frac{1+v}{2v} \rfloor}}$ can be expressed as $(c_1 + c_2 v)\left(\sqrt{c_3 + \frac{c_4}{v}} + \epsilon\right)^{\frac{2+v}{2v}}$. This suggests that there exists an optimal $v$ that achieves the minimal value. In Section 5.1, we empirically study the effects of $s$ across real-world datasets, and demonstrate that there exist specific ranges of $s$ that provide favorable trade-offs between delay and noise (Figure 6).

**Corollary 1.** *Let Assumptions 1 and 2 hold. Assume $F$ is a convex function. Ignoring the constants, the convergence rate under learning rate $\alpha^t = O\left(\frac{1}{\sqrt{t}}\right)$ simplifies to*

$$\min_{t \in [T]} \mathbb{E}[F(w^t)] - F(w^*) \leq O\left(\frac{1}{\sqrt{T}} \max_{t \in T_s} \mathbb{E}\left[\|D^t\|_1\right]\right) + O\left(\frac{1}{T} \sum_{t=1}^{T} \frac{1}{\sqrt{t}} \mathbb{E}\left[\|N^t\|_{D^t}^2\right]\right),$$

*where $T_s$ denotes the iteration indices where we switch from private RMSProp steps to private SGD steps plus the last iteration (thus having a constant cardinality) and $N^t \sim \mathcal{N}(\mathbf{0}, \sigma^2 C^2/b^2)$.*

At a high level, the first term is due to adaptive optimization using RMSProp, and the second term corresponds to the added privacy noise. Our $O\left(\frac{1}{\sqrt{T}}\right)$ rate is the same as previous results for SGD (or DP-SGD) in convex cases with delaying learning rates (Bassily et al., 2014; Nemirovski et al., 2009). Compared with DP-SGD, the added privacy noise would be reduced from $\frac{1}{T} \sum_{t=1}^{T} \frac{1}{\sqrt{t}} \mathbb{E}[\|N^t\|^2]$ to $\frac{1}{T} \sum_{t=1}^{T} \frac{1}{\sqrt{t}} \mathbb{E}[\|N^t\|_{D^t}^2]$ when the gradients are sparse (so that $\|D^t\|_1 < d$ in adaptive iterations). Hence, this theorem suggests some constant improvements relative to DP-SGD when we switch for a constant number of times.

## 4.2 Non-Convex Cases

We make the following additional common assumptions in non-convex convergence analyses.

**Assumption 3** (Smoothness). *Each $f(x^i; w)$ $(i \in [n])$ is $L$-smooth with respect to $w \in \mathbb{R}^d$.*

**Assumption 4.** *Stochastic gradient variance is bounded, i.e., $\mathbb{E}[\|g^{i,t} - \mathbb{E}[g^{i,t}]\|_2^2] \leq \tau^2$ for all $i, t$.*

**Theorem 3** (Convergence of Algorithm 1 for non-convex problems.)**.** *Let Assumptions 1-4 hold. Define constant $M$ as $M := C^2 + \frac{\sigma^2 C^2}{sb^2}$. Under any delay parameter $s$, after running Algorithm 1 with constant learning rates $\alpha^t = \alpha$ such that $\frac{L\alpha}{\epsilon} \leq 1$, we have*

$$\frac{1}{T}\sum_{t=1}^{T}\mathbb{E}[\|\nabla F(w^t)\|^2] \leq \frac{2(\sqrt{M}+1)F(w^1)}{\alpha T} + 2\alpha L(\sqrt{M}+1)\left(\frac{\tau^2}{2\epsilon^2 b} + \frac{d\sigma^2 C^2}{2b^2}\right).$$

The proof is deferred to Appendix B. Compared with Theorem 2, here we do not have constraints on $s$. Note that to guarantee $(\varepsilon, \delta)$-DP by running $T$ iterations, we can set $\sigma^2 = O\left(\frac{b^2 T \log(1/\delta)}{n^2 \varepsilon^2}\right)$, $\alpha = O\left(\frac{1}{\sqrt{d}}\right)$, and $T = O\left(\frac{n\varepsilon}{\log(1/\delta)}\right)$, to arrive at a convergence bound $O\left(\frac{\sqrt{d}}{n\varepsilon} + \frac{\tau^2}{\sqrt{db}}\right)$. Under any $s$, our rate (with and without noise) is the same as previous results on DP-SGD and (DP) adaptive methods for non-convex problems (Li et al., 2022; Zaheer et al., 2018). We note that our non-convex analysis does not directly highlight the benefits of adaptivity or trade-offs around $s$; hence the optimal choice of $s$ according to this result is $s = T$, to maximize the goal of reducing privacy noise. However, the practical performance can be better than the upper bound derived here, as shown in our experiments (Section 5). Most of the previous works studying stochastic non-convex adaptive optimization does not prove improvements relative to SGD (e.g., Alacaoglu et al., 2020; De et al., 2018; Ward et al., 2020; Zaheer et al., 2018). It is still an open problem to rigorously characterize the benefits of adaptivity for non-convex problems, which we leave for future work.

## 5 EMPIRICAL EVALUATION

In this section we report empirical results on a range of learning tasks. In Section 5.1, we compare $\mathrm{DP}^2$ with the baselines of DP-SGD and vanilla DP adaptive methods across various privacy budgets, and investigate the effects of delay on all datasets. We additionally compare $\mathrm{DP}^2$ with recent more advanced private adaptive methods in Section 5.2, and conduct ablation studies to validate the effectiveness of different $\mathrm{DP}^2$ components in Section 5.3.

In all experiments, we use Rényi differential privacy (RDP) accountant for the subsampled Gaussian mechanism (Mironov et al., 2019) for privacy accounting. We focus on the RMSProp optimizer (Hinton et al., 2012) and provide results relating to other adaptive methods such as AdaGrad (Duchi et al., 2011; Streeter & McMahan, 2010) in Appendix C. Our experiments are implemented in JAX (Bradbury et al., 2018) with Haiku (Hennigan et al., 2020) to auto-vectorize over the per-example operations (e.g. per-example clipping) for substantial speedups (Subramani et al., 2021). Unless explicitly stated, we report results with the best grid-searched hyperparameters. Note that for $\mathrm{DP}^2$ we tune the learning rates and clipping thresholds separately for private SGD iterations and private adaptive (RMSProp) iterations. See Appendix C.2 for hyperparameter details. Our code is publicly available at `github.com/kenziyuliu/DP2`.

**Tuning $s$.** In all experiments, we tune the delay parameter ($s$) via grid search. For convex tasks, we choose $s$ from $\{0.025, 0.5, 0.1, 0.5, 1, 2\}$ epochs. For the non-convex model, we choose $s$ from $\{0.5, 3, 10, 25\}$ epochs. We explore the sensitivity of $\mathrm{DP}^2$ to $s$ in Section 5.2, and show that there exist a wide range of $s$ parameters that result in superior performance compared with baseline methods.

**Datasets and Tasks.** We pick datasets and tasks where adaptivity is crucial (e.g., those involving sparse gradients). For such tasks, adaptive methods have major benefits relative to SGD in non-private training, and we expect $\mathrm{DP}^2$ to retain the benefits in private training. See Appendix C.1 for a detailed description. For all datasets, we explore the effects of several noise multiplier ($\sigma$) values, and set $\delta = 10^{-k}$ where $k$ is the smallest integer that satisfies $10^{-k} \leq 1/n$ for the training dataset size $n$.

### 5.1 $\mathrm{DP}^2$ COMPARED WITH DP-SGD AND VANILLA DP ADAPTIVE METHODS

We consider two popular baselines: DP-SGD (Abadi et al., 2016) and vanilla DP-RMSProp (Zhou et al., 2020). In vanilla DP adaptive methods, private gradients are plugged into adaptive updating rules to approximate the preconditioners at each iteration. Figure 4 compares $\mathrm{DP}^2$-RMSProp with DP-SGD and DP-RMSProp. We observe that across all datasets, $\mathrm{DP}^2$ consistently and substantially outperforms the baselines in terms of both convergence and absolute performance.

**Privacy/utility trade-offs.** Figure 4 reports learning curves under specific privacy budgets determined by the batch size and the number of epochs. Here, we additionally explore privacy/utility trade-offs

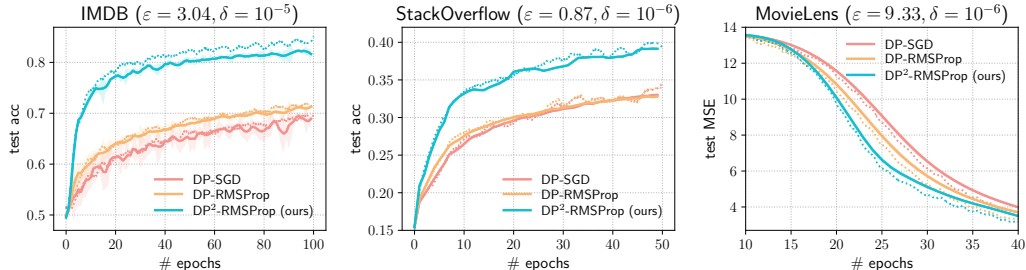

Figure 4: **Test performance** of DP$^2$ compared to DP-SGD and DP-RMSProp on IMDB (left), StackOverflow (middle), and MovieLens-100k (right) for a fixed privacy budget. For all datasets, we calculate the privacy loss ($\varepsilon$) under fixed $\delta$'s, noise multipliers $\{1.0, 1.0, 0.5\}$, and batch size 64. All runs are repeated over 5 random seeds. Dotted lines correspond to training metrics.

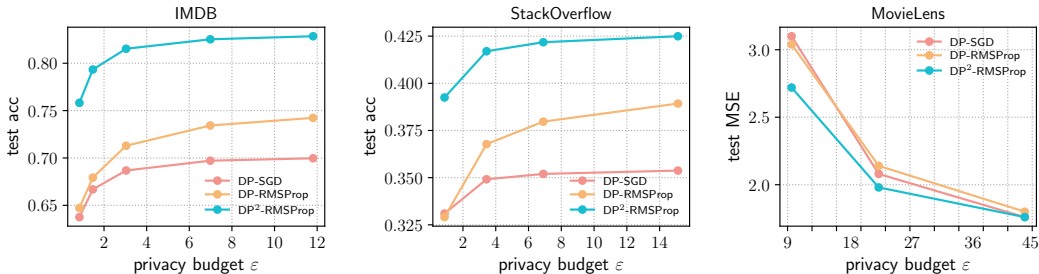

Figure 5: **Privacy/utility trade-offs** of DP$^2$-RMSProp (Algorithm 1) compared with DP-SGD and DP-RMSProp for a range of privacy budgets. We see that DP$^2$-RMSProp consistently achieves more favorable privacy/utility trade-offs than the baseline methods.

across a range of privacy parameters, where $\varepsilon$ ranges are consistent with prior works (e.g., Kairouz et al., 2021b). Results are shown in Figure 5. We observe that similar to the results in Figure 4, DP$^2$ significantly outperforms DP-SGD and DP-RMSProp under each privacy budget. For reference, the non-private RMSProp method achieves 87% accuracy, 62% accuracy, and 0.88 mean square error (MSE) on IMDB, StackOverflow, and MovieLens, respectively. Indeed, with weaker privacy (larger $\varepsilon$), we expect smaller utility gaps between private and non-private optimization. In Appendix C.4, we additionally explore how increasing the computational budget may affect the privacy-utility trade-off.

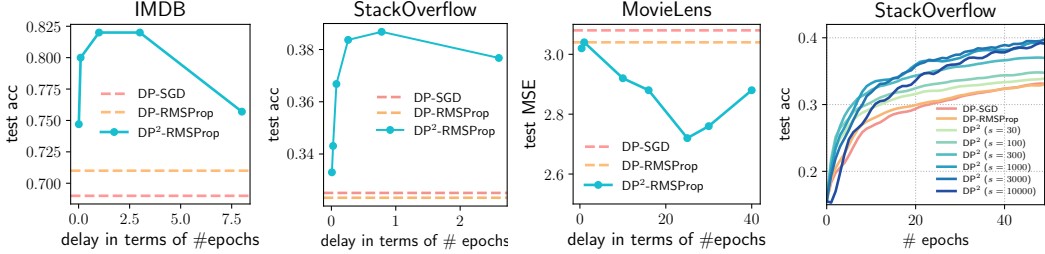

Figure 6: **Effect of the delay parameter** $s$. We show trade-offs between delay and noise in the first three subplots. The rightmost subfigure showcases convergence curves under different delays ($s=10000$ corresponds to delaying for $\approx 3$ epochs) where DP$^2$ achieves $4\times$ convergence speedup than DP-SGD. Privacy settings follow those of Figure 4. Although a specific value of $s$ achieves the greatest improvements, we observe that nearly all instantiations of DP$^2$ improve upon the baselines.

**Effects of** $s$**.** Finally, we empirically study the effect of the delay parameter $s$. Intuitively, there exists a trade-off between the amount of delay and the privacy noise in the preconditioner: averaging over more historical gradients (larger $s$) could yield less noisy preconditioners, while introducing more staleness. In Figure 6, we report test performance versus the delay $s$ across all datasets on the first three subplots. In the last subplot, we additionally show the convergence behavior under different values of $s$. These results suggest that there is a "sweet spot" for $s$ to yield good performance— small delays are gradually improving over DP-RMSProp; moderate delays perform best in terms of convergence and absolute performance; and large delays may slow down convergence (although it is

possible to reach similar performance with sufficient training). These empirical results are consistent with the implications of our convergence analysis discussed in Section 4.1.

## 5.2 DP² COMPARED WITH RECENT METHODS FOR PRIVATE OPTIMIZATION

As discussed in Section 2, beyond DP-SGD and vanilla DP adaptive methods, another line of work uses *auxiliary, public data* to improve private (adaptive) optimization. While *not directly comparable* to DP² since DP² does not require any side/public information, we compare DP² to two state-of-the-art methods along this direction[2]: (1) AdaDPS (Li et al., 2022) which uses public data or their statistics to estimate gradient geometry, and (2) PDA-DPMD (Amid et al., 2022), which uses the loss on public data as a mirror map to learn the underlying gradient geometry. Results are reported in Table 1, which show that DP² has comparable performance to state-of-the-art baselines, but without the need to access auxiliary data. See Appendix C.6 for full details and convergence curves.

| Dataset | DP-SGD | DP-RMSProp | PDA-DPMD | AdaDPS (w/ RMSProp) | DP²-RMSProp |
|---|---|---|---|---|---|
| IMDB ↑ | $.687 \pm .018$ | $.713 \pm .005$ | $.703 \pm .005$ | $\mathbf{.826} \pm .003$ | $\mathbf{.815} \pm .011$ |
| StackOverflow ↑ | $.330 \pm .002$ | $.328 \pm .002$ | $.353 \pm .001$ | $\mathbf{.406} \pm .027$ | $\mathbf{.391} \pm .001$ |
| MovieLens ↓ | $3.02 \pm .068$ | $2.96 \pm .062$ | $3.74 \pm .053$ | $2.86 \pm .042$ | $\mathbf{2.78} \pm .054$ |

Table 1: DP² compared with other private (adaptive) methods that use public data (Amid et al., 2022; Li et al., 2022). Even though DP² *does not* require auxiliary information, we find that it achieves comparable performance with these state-of-the-art approaches that require additional public data. Corresponding convergence plots are presented in Figure 11 in Appendix C.6.

## 5.3 ABLATION STUDIES

Finally, we also study the effectiveness of different components of DP². Recall that in Algorithm 1, we use noisy gradients from DP-SGD iterations to update both the model parameters and the preconditioner such that the total privacy cost is identical to that of DP-SGD. The first variant considers accumulating DP-SGD gradients in the same way, but it runs private adaptive methods using delayed preconditioner in almost all iterations. This requires us to add independent noise *twice* at most iterations (when accumulating the preconditioner and when noising the preconditioned update), thus increasing the total privacy budget. The second variant is identical to DP² except that it applies the delayed preconditioner *after* noising the clean gradient; this is to study the order of preconditioning as discussed in Section 3. As illustrated in Figure 7, both

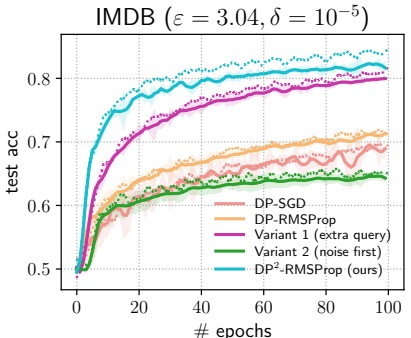

Figure 7: Different ablation variants of DP² on IMDB. The dotted lines correspond to training accuracy.

variants indeed significantly underperform our proposed method on the IMDB dataset, thus validating the design choices of DP². We defer complete results to Figure 10 and Table 4 in Appendix C.5. See also Appendix D for the exact algorithms of both variants.

## 6 CONCLUSION AND FUTURE WORK

In this work, we proposed DP², a private adaptive optimization framework that uses historical gradients to construct delayed but less noisy preconditioners, yielding improved privacy/utility trade-offs *without the need to access auxiliary data*. We demonstrated the effectiveness of DP² both theoretically and empirically. In the future, it would be interesting to extend the techniques developed herein to other privacy-sensitive applications such as federated learning (McMahan et al., 2017; Reddi et al., 2021). It is also worth exploring interplays between DP² and private online optimization with tree aggregation, which similarly releases cumulative statistics with reduced noise (Chan et al., 2011).

---

[2]We do not directly compare with the prior work of Asi et al. (2021) as the code is not publicly available and implementation details are missing in the paper; however, the more recent PDA-DPMD work of Amid et al. (2022) we compare with suggests superior performance to Asi et al. (2021). We also implement the diagonal variant of the method proposed in the theoretically-focused work of Kairouz et al. (2021a), but observe that accuracy improves only marginally beyond random guessing (see Figure 12 in Appendix C.6).

## ACKNOWLEDGMENTS

The work of TL, ZL, and VS was supported in part by the National Science Foundation Grant IIS1838017, a Google Faculty Award, a Meta Faculty Award, the Private AI Collaborative Research Institute, and the CONIX Research Center. Any opinions, findings, and conclusions or recommendations expressed in this material are those of the author(s) and do not necessarily reflect the National Science Foundation or any other funding agency.

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

## A  PROOFS

**Lemma 1.** *Under Assumption 2, let $s_1 = s_2 = s$ in Algorithm 1, we have for any $j \in [d]$, $\mathbb{E}[v_j] \le C^2 + \frac{\sigma^2 C^2}{sb^2}$.*

*Proof.* Recall that $C$ is the gradient norm bound (Assumption 2). Let the clipping threshold be $C$ as well. We have for $j \in [d]$,

$$\mathbb{E}\left[\left(\frac{1}{s}G_j\right)^2\right] = \mathbb{E}\left[\left(\frac{1}{s}\left(g_j^{i_1} + \cdots + g_j^{i_s}\right) + \frac{1}{s}\left(N_j^{i_1} + \cdots + N_j^{i_s}\right)\right)^2\right] \tag{1}$$

$$= \mathbb{E}\left[\frac{1}{s^2}\left(g_j^{i_1} + \cdots + g_j^{i_s}\right)^2\right] + \mathbb{E}\left[\frac{1}{s^2}\left(N_j^{i_1} + \cdots + N_j^{i_s}\right)^2\right] \tag{2}$$

$$\le C^2 + \frac{\sigma^2 C^2}{sb^2}, \tag{3}$$

where $\{i_1, \ldots, i_s\}$ denotes the indices of $s$ noisy gradients used to obtain $G_j$, and $\{N_j^{i_1}, \ldots, N_j^{i_s}\}$ are random zero-mean Gaussian variables with variance $\frac{\sigma^2 C^2}{b^2}$ under noise multiplier $\sigma$, clipping threshold $C$, and mini-batch size $b$. Hence for any $j \in [d]$ and $t \in [T]$,

$$\mathbb{E}\left[\left(\frac{1}{s}G_j\right)^2\right] \le C^2 + \frac{\sigma^2 C^2}{sb^2} := M, \tag{4}$$

$$\mathbb{E}[v_j] \le M, \tag{5}$$

$$\mathbb{E}\left[\sqrt{v_j}\right] \le \sqrt{\mathbb{E}[v_j]} \le \sqrt{M} \tag{6}$$

$$\mathbb{E}\left[D_j^t\right] \le \max\left\{\sqrt{M} + \epsilon, 1\right\}. \tag{7}$$

$\square$

### A.1  PROOF OF THEOREM 2

Based on the updating rule, we have

$$\left\|w^{t+1} - w^*\right\|_{D^t}^2 \tag{8}$$

$$= \left\|w^t - \alpha^t \frac{g^t}{D^t} - \alpha^t N^t - w^*\right\|_{D^t}^2 \tag{9}$$

$$= \left\|w^t - w^*\right\|_{D^t}^2 + \left\|\alpha^t \frac{g^t}{D^t} + \alpha^t N^t\right\|_{D^t}^2 - 2\left\langle w^t - w^*, \alpha^t g^t + \alpha^t D^t N^t\right\rangle \tag{10}$$

$$= \left\|w^t - w^*\right\|_{D^t}^2 - 2\alpha^t\left\langle g^t, w^t - w^*\right\rangle + (\alpha^t)^2\left\langle g^t, \frac{g^t}{D^t}\right\rangle$$
$$- 2\alpha^t\left\langle w^t - w^*, D^t N^t\right\rangle + (\alpha^t)^2\|N^t\|_{D^t}^2 + 2(\alpha^t)^2\left\langle g^t, N^t\right\rangle. \tag{11}$$

Rearranging terms gives

$$\left\langle g^t, w^t - w^*\right\rangle = \frac{\|w^t - w^*\|_{D^t}^2 - \|w^{t+1} - w^*\|_{D^t}^2}{2\alpha^t} + \frac{\alpha^t}{2}\left\langle g^t, \frac{g^t}{D^t}\right\rangle$$
$$- \left\langle w^t - w^*, D^t N^t\right\rangle + \frac{\alpha^t}{2}\|N^t\|_{D^t}^2 + \alpha^t\left\langle g^t, N^t\right\rangle. \tag{12}$$

Taking the expectation on both sides conditioned on $w^t$,

$$\left\langle \nabla F(w^t), w^t - w^*\right\rangle = \frac{\mathbb{E}_t\left[\|w^t - w^*\|_{D^t}^2\right] - \mathbb{E}_t[\|w^{t+1} - w^*\|_{D^t}^2]}{2\alpha^t}$$
$$+ \frac{\alpha^t}{2}\mathbb{E}_t\left[\left\langle g^t, \frac{g^t}{D^t}\right\rangle\right] + \frac{\alpha^t}{2}\mathbb{E}_t\left[\|N^t\|_{D^t}^2\right], \tag{13}$$

where we have used the fact that $N$ is a zero-mean Gaussian variable independent of $g^t, w^t$. Taking the expectation on both sides and using the convexity of $F(\cdot)$:

$$\mathbb{E}[F(w^t)] - F(w^*)$$
$$\leq \frac{\mathbb{E}[\|w^t - w^*\|^2_{D^t}] - \mathbb{E}[\|w^{t+1} - w^*\|^2_{D^t}]}{2\alpha^t} + \frac{\alpha^t}{2}\mathbb{E}\left[\left\langle g^t, \frac{g^t}{D^t}\right\rangle\right] + \frac{\alpha^t}{2}\mathbb{E}\left[\|N^t\|^2_{D^t}\right]. \quad (14)$$

Applying telescope sum, we have

$$\sum_{t=1}^{T}\left(\mathbb{E}[F(w^t)] - F(w^*)\right)$$
$$\leq \frac{\|w^1 - w^*\|^2_{A^1}}{2\alpha_1} + \sum_{t=2}^{T}\left(\frac{\mathbb{E}\left[\|w^t - w^*\|^2_{D^t}\right]}{2\alpha^t} - \frac{\mathbb{E}\left[\|w^t - w^*\|^2_{D^{t-1}}\right]}{2\alpha_{t-1}}\right)$$
$$+ \sum_{t=1}^{T}\frac{\alpha^t}{2}\mathbb{E}\left[\left\langle g^t, \frac{g^t}{D^t}\right\rangle\right] + \sum_{t=1}^{T}\frac{\alpha^t}{2}\mathbb{E}\left[\|N^t\|^2_{D^t}\right]. \quad (15)$$

Hence, we need to bound the RHS:

$$\frac{\|w^1 - w^*\|^2_{D^1}}{2\alpha^2} + \underbrace{\sum_{t=2}^{T}\left(\frac{\mathbb{E}\left[\|w^t - w^*\|^2_{D^t}\right]}{2\alpha^t} - \frac{\mathbb{E}\left[\|w^t - w^*\|^2_{D^{t-1}}\right]}{2\alpha^{t-1}}\right)}_{T_1}$$
$$+ \underbrace{\sum_{t=1}^{T}\frac{\alpha^t}{2}\mathbb{E}\left[\left\langle g^t, \frac{g^t}{D^t}\right\rangle\right]}_{T_2} + \sum_{t=1}^{T}\frac{\alpha^t}{2}\mathbb{E}\left[\|N^t\|^2_{D^t}\right], \quad (16)$$

where the vector $D^t \in \mathbb{R}^d$ satisfies that $D^t = \mathbf{1}$ when running private SGD steps, and $D^t = \sqrt{v} + \epsilon$ when running private RMSProp steps.

Let the delay parameter to be scheduled as

$$s = \upsilon T \ (0 < \upsilon < 1) \quad (17)$$

and the learning rate $\alpha^t$ be

$$\alpha^t \leftarrow \frac{\alpha^{\lfloor\frac{t}{2s}\rfloor + \lfloor\frac{t+s}{2s}\rfloor + 1}}{\sqrt{t}}, \quad (18)$$

where $\alpha = \min\left\{\epsilon, \frac{1}{\sqrt{M}+\epsilon}, 1\right\}$, and $M$ is the upper bound of $\mathbb{E}[v_j]$ for $j \in [d]$, as defined and proved in Lemma 1.

We next consider the $T_1$ term. There are four cases.

1. **DP-SGD at the $t-1$-th iteration, and DP-SGD at the $t$-th iteration**: As $D^t = D^{t-1}$ there is not much requirement other that the learning rates need to satisfy $\alpha^t \leq \alpha^{t-1}$, which holds for our choice.

2. **Private RMSProp at the $t-1$-th iteration, and private RMSProp at the $t$-th iteration**: Similar to previous case, the learning rates need to satisfy $\alpha^t \leq \alpha^{t-1}$, which holds for our choice.

3. **DP-SGD at the $t-1$-th iteration, and private RMSProp at the $t$-th iteration**: We require

$$\frac{\alpha^t}{\epsilon} \leq \alpha^{t-1} \implies \frac{\sqrt{v^t}+\epsilon}{\alpha^t} \geq \frac{1}{\alpha^{t-1}} \quad (19)$$

   But in this case we must have $t \ \% \ s = 0$. So this is satisfied by our choice as long as $\alpha \leq \epsilon$.

4. **Private RMSProp at the $t-1$-th iteration, and DP-SGD at the $t$-th iteration**

The first three cases form an updating pattern of DP-SGD $\rightarrow \cdots \rightarrow$ DP-SGD $\rightarrow$ DP-RMSProp$\rightarrow$ $\cdots \rightarrow$ DP-RMSProp, where every pattern takes $2s$ iterations, except for the first pattern, because the telescope sum starts from $t = 2$. For the first pattern, we have

$$\frac{\left\|w^1 - w^*\right\|_{D^1}^2}{2\alpha^2} + \sum_{t=2}^{2s} \left(\frac{\mathbb{E}\left[\|w^t - w^*\|_{D^t}^2\right]}{2\alpha^t} - \frac{\mathbb{E}\left[\|w^t - w^*\|_{D^{t-1}}^2\right]}{2\alpha^{t-1}}\right) \tag{20}$$

$$= \frac{\left\|w^1 - w^*\right\|_{D^1}^2}{2\alpha^2} + \sum_{t=2}^{2s} \left(\mathbb{E}\left[\|w^t - w^*\|_{\frac{D^t}{\alpha^t} - \frac{D^{t-1}}{\alpha^{t-1}}}^2\right]\right) \tag{21}$$

$$\leq \frac{\left\|w^1 - w^*\right\|_{D^1}^2}{2\alpha^2} + R^2 \sum_{t=2}^{2s} \left(\frac{\mathbb{E}\left[\|D^t\|_1\right]}{2\alpha^t} - \frac{\mathbb{E}\left[\|D^{t-1}\|_1\right]}{2\alpha^{t-1}}\right) \leq \frac{R^2}{2\alpha^{2s}} \mathbb{E}\left[\|D^{2s}\|_1\right], \tag{22}$$

where $D^{2s} = \sqrt{v} + \epsilon$.

For $k \geq 1$, we have

$$\sum_{t=2sk+1}^{2sk+2s} \left(\frac{\mathbb{E}\left[\|w^t - w^*\|_{D^t}^2\right]}{2\alpha^t} - \frac{\mathbb{E}\left[\|w^t - w^*\|_{D^{t-1}}^2\right]}{2\alpha^{t-1}}\right)$$

$$= \frac{\mathbb{E}\left[\|w^{2sk+1} - w^*\|_{D^{2sk+1}}^2\right]}{2\alpha^{2sk+1}} - \frac{\mathbb{E}\left[\|w^{2sk+1} - w^*\|_{D^{2sk}}^2\right]}{2\alpha^{2sk}} + \sum_{t=2sk+2}^{2sk+2s} \left(\mathbb{E}\left[\|w^t - w^*\|_{\frac{D^t}{2\alpha^t} - \frac{D^{t-1}}{2\alpha^{t-1}}}^2\right]\right)$$

$$\leq \frac{\mathbb{E}\left[\|w^{2sk+1} - w^*\|_{D^{2sk+1}}^2\right]}{2\alpha^{2sk+1}} - \frac{\mathbb{E}\left[\|w^{2sk+1} - w^*\|_{D^{2sk}}^2\right]}{2\alpha^{2sk}} + R^2 \left(\frac{\mathbb{E}[\|D^{2sk+2s}\|_1]}{2\alpha^{2sk+2s}} - \frac{\mathbb{E}[\|D^{2sk+1}\|_1]}{2\alpha^{2sk+1}}\right)$$

$$\leq \frac{\mathbb{E}\left[\|w^{2sk+1} - w^*\|_{D^{2sk+1}}^2\right]}{2\alpha^{2sk+1}} + R^2 \left(\frac{\mathbb{E}[\|D^{2sk+2s}\|_1]}{2\alpha^{2sk+2s}} - \frac{\mathbb{E}[\|D^{2sk+1}\|_1]}{2\alpha^{2sk+1}}\right)$$

$$\leq \frac{R^2}{2\alpha^{2sk+2s}} \mathbb{E}\left[\|D^{2sk+2s}\|_1\right], \tag{23}$$

where $D^{2sk+2s} = \sqrt{v} + \epsilon$ belong to DP-RMSProp updates.

We look at the second $T_2$ term, and prove by induction that there exists a constant $\kappa$ such that

$$\sum_{t=1}^{T} \frac{\alpha^t}{2} \mathbb{E}\left[\left\langle g^t, \frac{g^t}{D^t}\right\rangle\right] \leq \frac{\kappa}{\alpha^T} \mathbb{E}\left[\|D^T\|_1\right]. \tag{24}$$

When $T = 1$ ($\alpha^1 = \alpha$ and $D^1 = \mathbf{1}$), $\frac{\alpha}{2}\mathbb{E}[\|g^1\|^2] \leq \frac{\kappa d}{\alpha}$ holds if $\kappa \geq \alpha^2 C^2$. At each step $t$, the goal is to get

$$\frac{\kappa}{\alpha^{t-1}} \mathbb{E}\left[\|D^{t-1}\|_1\right] + \frac{\alpha^t}{2} \mathbb{E}\left[\left\langle g^t, \frac{g^t}{D^t}\right\rangle\right] \leq \frac{\kappa}{\alpha^t} \mathbb{E}\left[\|D^t\|_1\right] \tag{25}$$

1. **DP-SGD at the $t - 1$-th iteration, and DP-SGD at the $t$-th iteration**: We require

$$\frac{\kappa d}{\alpha^{t-1}} + \frac{\alpha^t}{2} \mathbb{E}\left[\|g^t\|^2\right] \leq \frac{\kappa d}{\alpha^t} \tag{26}$$

which would hold for choice of $\alpha^t$ as gradients are bounded and $\kappa \geq \alpha^2 C^2$.

2. **Private RMSProp at the $t - 1$-th iteration, and private RMSProp at the $t$-th iteration**: We need

$$\frac{\kappa \mathbb{E}\left[\|\sqrt{v^{t-1}} + \epsilon\|_1\right]}{\alpha^{t-1}} + \frac{\alpha^t}{2} \mathbb{E}\left[\left\langle g^t, \frac{g^t}{\sqrt{v^{t-1}} + \epsilon}\right\rangle\right] \leq \frac{\kappa}{\alpha^t} \mathbb{E}\left[\|\sqrt{v^t} + \epsilon\|_1\right], \tag{27}$$

$$\frac{\alpha^t}{2} \mathbb{E}\left[\left\langle g^t, \frac{g^t}{\sqrt{v^{t-1}} + \epsilon}\right\rangle\right] \leq \left(\frac{\kappa}{\alpha^t} - \frac{\kappa}{\alpha^{t-1}}\right) \mathbb{E}\left[\left\|\sqrt{v^{t-1}} + \epsilon\right\|_1\right]. \tag{28}$$

Let

$$h(s) \geq \max_{t \in [T]} \left\{ \frac{\mathbb{E}\left[\|g^t\|_1\right]}{\mathbb{E}\left[\left\|\frac{1}{s}\left|G^{\lfloor \frac{t}{s}\rfloor s}\right| + \epsilon\right\|_1\right]} \right\}. \tag{29}$$

Based on our updating rule,

$$\mathbb{E}\left[\left\|\sqrt{v^t} + \epsilon\right\|_1\right] \geq \sqrt{1-\beta}\,\mathbb{E}\left[\left\|\frac{1}{s}\left|G^{\lfloor \frac{t}{s}\rfloor s}\right| + \epsilon\right\|_1\right]. \tag{30}$$

Note that

$$\frac{\alpha^t}{2}\mathbb{E}\left[\left\langle g^t, \frac{g^t}{\sqrt{v^{t-1}}+\epsilon}\right\rangle\right] \leq \frac{\alpha^t}{2}\mathbb{E}\left[\frac{\|g^t\|^2}{\epsilon}\right] \leq \frac{\alpha^t C}{2\epsilon}\mathbb{E}[\|g^t\|] \leq \frac{\alpha^t C}{2\epsilon}\mathbb{E}[\|g^t\|_1], \tag{31}$$

where we have used the assumption that $\|g^t\| \leq C$. Combining the above two,

$$\frac{\alpha^t C}{2\epsilon}\mathbb{E}[\|g^t\|] \leq \frac{\alpha^t C}{2\epsilon}h(s)\mathbb{E}\left[\left\|\frac{1}{s}\left|G^{\lfloor \frac{t}{s}\rfloor s}\right| + \epsilon\right\|_1\right] \tag{32}$$

$$\leq \frac{\alpha^t C}{2\epsilon}\frac{h(s)}{\sqrt{1-\beta}}\mathbb{E}\left[\left\|\sqrt{v^{t-1}} + \epsilon\right\|_1\right] \tag{33}$$

$$\leq \kappa\left(\frac{1}{\alpha^t} - \frac{1}{\alpha^{t-1}}\right)\mathbb{E}\left[\left\|\sqrt{v^{t-1}} + \epsilon\right\|_1\right]. \tag{34}$$

This implies the condition holds as long as $\kappa$ satisfies

$$\kappa \geq \frac{Ch(s)}{\epsilon\sqrt{1-\beta}}. \tag{35}$$

3. **DP-SGD at the $t-1$-th iteration, and private RMSProp at the $t$-th iteration.** We want to prove

$$\frac{\kappa d}{\alpha^{t-1}} + \frac{\alpha^t}{2}\mathbb{E}\left[\left\langle g^t, \frac{g^t}{D^t}\right\rangle\right] \leq \frac{\kappa}{\alpha^t}\mathbb{E}\left[\|D^t\|_1\right]. \tag{36}$$

As $\|g^t\| \leq C$, it holds that

$$\frac{\alpha^t}{2}\mathbb{E}\left[\left\langle g^t, \frac{g^t}{\sqrt{v^t}+\epsilon}\right\rangle\right] \leq \frac{\alpha^t}{2\epsilon}\mathbb{E}[\|g^t\|^2] \leq \frac{\alpha^t C}{2\epsilon}\mathbb{E}[\|g^t\|] \leq \frac{\alpha^t C}{2\epsilon}\mathbb{E}[\|g^t\|_1]. \tag{37}$$

Therefore,

$$\frac{\alpha^t}{2}\mathbb{E}\left[\left\langle g^t, \frac{g^t}{\sqrt{v^t}+\epsilon}\right\rangle\right] \leq \frac{Ch(s)}{2\epsilon\sqrt{1-\beta}}\alpha^t\mathbb{E}\left[\left\|\sqrt{v^t} + \epsilon\right\|_1\right]. \tag{38}$$

Based on our learning rate set in Eq. (18),

$$\sqrt{t}\alpha^t = \sqrt{t-1}\alpha^{t-1}\epsilon \tag{39}$$

$$\implies \frac{\alpha^t}{2} \leq \frac{1}{\alpha^t} - \frac{1}{\alpha^{t-1}\epsilon} \leq \frac{1}{\alpha^t} - \frac{d}{\alpha^{t-1}\mathbb{E}\left[\|D^t\|_1\right]}. \tag{40}$$

Hence,

$$\frac{Ch(s)}{2\epsilon\sqrt{1-\beta}}\alpha^t\mathbb{E}\left[\left\|\sqrt{v^t} + \epsilon\right\|_1\right] \leq \frac{Ch(s)}{\epsilon\sqrt{1-\beta}}\mathbb{E}\left[\|D^t\|_1\right]\left(\frac{1}{\alpha^t} - \frac{d}{\alpha^{t-1}\mathbb{E}\left[\|D^t\|_1\right]}\right) \tag{41}$$

$$\leq \kappa\left(\frac{\mathbb{E}[\|D^t\|_1]}{\alpha^t} - \frac{d}{\alpha^{t-1}}\right), \tag{42}$$

where we require

$$\kappa \geq \frac{Ch(s)}{\epsilon\sqrt{1-\beta}}. \tag{43}$$

4. **Private RMSProp at the $t-1$-th iteration, and DP-SGD at the $t$-th iteration.** We need

$$\frac{\kappa}{\alpha^{t-1}}\mathbb{E}\left[\left\|\sqrt{v^{t-1}}+\epsilon^{t-1}\right\|_1\right] + \frac{\alpha^t}{2}\mathbb{E}\left[\|g^t\|^2\right] \le \frac{\kappa d}{\alpha^t}. \tag{44}$$

Plug in $\mathbb{E}\left[\|\sqrt{v^{t-1}}\|_1\right] \le d\sqrt{M}$ (Lemma 1) and $\|g^t\|^2 \le C^2$, we have

$$\frac{\kappa}{\alpha^{t-1}}\mathbb{E}\left[\|\sqrt{v^{t-1}}+\epsilon\|_1\right] + \frac{\alpha^t}{2}\mathbb{E}\left[\|g^t\|^2\right] \le \frac{\kappa}{\alpha^{t-1}}\left(d\sqrt{M}+d\right) + \frac{\alpha^t}{2}C^2. \tag{45}$$

Based on our learning rate set in Eq. (18), for some constant $\gamma$,

$$\alpha^{t-1} = \frac{\gamma}{\sqrt{t-1}}, \ \alpha^t \le \frac{\gamma}{\sqrt{t}(\sqrt{M}+1)} \tag{46}$$

$$\implies \frac{\alpha^t}{2} \le \frac{1}{\alpha^t} - \frac{\sqrt{M}+1}{\alpha^{t-1}} \le \frac{d}{\alpha^t} - \frac{d\sqrt{M}+d}{\alpha^{t-1}}. \tag{47}$$

Therefore

$$\frac{\alpha^t}{2}C^2 \le \kappa\left(\frac{d}{\alpha^t} - \frac{d\sqrt{M}+d}{\alpha^{t-1}}\right) \tag{48}$$

holds as long as $\kappa \ge \alpha^2 C^2$. To sum up, the requirement on $\kappa$ is

$$\kappa \ge \max\left\{\alpha^2 C^2, \frac{Ch(s)}{\epsilon\sqrt{1-\beta}}\right\}. \tag{49}$$

Final convergence results:

$$\min_{t\in[T]}\mathbb{E}\left[F(w^t)\right] - F(w^*) \tag{50}$$

$$\le \frac{R^2+\kappa}{\alpha^{\lfloor\frac{1}{2v}\rfloor+\lfloor\frac{1+v}{2v}\rfloor}}\frac{1}{\sqrt{T}}\sum_{t\in T_v}\mathbb{E}\left[\|D^t\|_1\right] + \frac{1}{T}\sum_{t=1}^{T}\frac{\alpha^{\lfloor\frac{t}{2vT}\rfloor+\lfloor\frac{t+vT}{2vT}\rfloor}}{\sqrt{t}}\mathbb{E}[\|N^t\|_{D^t}^2], \tag{51}$$

where $T_v$ denotes the iteration indices where we switch from private RMSProp steps to private SGD steps plus the last iteration, and its cardinality is $|T_v| = \lceil\frac{1}{2v}\rceil$, and $\kappa \ge \max\left\{\alpha^2 C^2, \frac{Ch(s)}{\epsilon\sqrt{1-\beta}},\right\}$, $\alpha = \min\left\{\epsilon, \frac{1}{\sqrt{M}+\epsilon}, 1\right\}$.

## A.2 A CLOSER LOOK AT $h(s)$

We closely examine $h(s)$, defined as

$$h(s) \ge \max_{t\in[T]}\left\{\frac{\mathbb{E}\left[\|g^t\|_1\right]}{\mathbb{E}\left[\left\|\frac{1}{s}\left|G^{\lfloor\frac{t}{s}\rfloor s}\right| + \epsilon\right\|_1\right]}\right\}. \tag{52}$$

Let us assume mini-batch gradients on consecutive time steps are not very different, i.e. $\|g^t - g^{t-1}\|_1 \le M$. This means each gradient norm cannot be too far away from each other, which can be used to show the dependence of $h(s)$ on the delay parameter $s$. Denote the gap between the current iteration $t$ and the iteration where $v$ gets updated as $k$, i.e., $k := t - \lfloor\frac{t}{s}\rfloor s$. Hence,

$$\frac{\|g^t\|_1}{\left\|\frac{1}{s}\left(g^{t-k-1}+\cdots+g^{t-k-s}\right) + \frac{1}{s}\left(N^{t-k-1}+\cdots+N^{t-k-s}\right)\right\|_1 + d\epsilon} \tag{53}$$

$$= \frac{\left\|g^t - \frac{1}{s}\left(g_j^{t-k-1}+\cdots+g_j^{t-k-s}\right) + \frac{1}{s}\left(g_j^{t-k-1}+\cdots+g_j^{t-k-s}\right)\right\|_1}{\left\|\frac{1}{s}\left(g_j^{t-k-1}+\cdots+g_j^{t-k-s}\right) + \frac{1}{s}\left(N^{t-k-1}+\cdots+N^{t-k-s}\right)\right\|_1 + d\epsilon} \tag{54}$$

$$= \frac{\left\|\frac{1}{s}\left((g^t-g^{t-k-1})+\cdots+(g^t-g^{t-k-s})\right) + \frac{1}{s}\left(g^{t-k-1}+\cdots+g^{t-k-s}\right)\right\|_1}{\left\|\frac{1}{s}\left(g^{t-k-1}+\cdots+g^{t-k-s}\right) + \frac{1}{s}\left(N^{t-k-1}+\cdots+N^{t-k-s}\right)\right\|_1 + d\epsilon} \tag{55}$$

$$\le \frac{\left\|\frac{1}{s}\left(g^{t-k-1}+\cdots+g^{t-k-s}\right)\right\|_1}{\left\|\frac{1}{s}\left(g^{t-k-1}+\cdots+g^{t-k-s}\right) + \frac{1}{s}\left(N^{t-k-1}+\cdots+N^{t-k-s}\right)\right\|_1 + d\epsilon} + \frac{\frac{1}{s}(sM+\cdots+(2s)M)}{d\epsilon} \tag{56}$$

Denote $a := \frac{1}{s} \left( N^{t-k-1} + \cdots + N^{t-k-s} \right)$, and $b := \frac{1}{s} \left( g^{t-k-1} + \cdots + g^{t-k-s} \right)$. Then

$$h(s) \leq \frac{\mathbb{E}[\|b\|_1]}{\mathbb{E}[\|a+b\|_1] + d\epsilon} + \frac{sM}{d\epsilon} \tag{57}$$

$$\leq \frac{1}{\left| \frac{\mathbb{E}[\|a\|_1]}{\mathbb{E}[\|b\|_1]} - 1 \right| + \frac{d\epsilon}{\mathbb{E}[\|b\|_1]}} + \frac{sM}{d\epsilon} \tag{58}$$

In the special case where gradients are sparse, i.e., $\mathbb{E}[\|b\|_1] < \mathbb{E}[\|a\|_1]$, we have

$$h(s) \leq \frac{1}{\frac{\mathbb{E}[\|a\|_1]}{\mathbb{E}[\|b\|_1]} + \frac{d\epsilon}{\mathbb{E}[\|b\|_1]} - 1} + \frac{sM}{d\epsilon} \tag{59}$$

It is easy to see that the RHS is $O(s)$, and it increases as $s$. We can informally express it as $c_1 s + c_2$, where $c_1$ and $c_2$ are two constants.

## B  PROOF OF THEOREM 3

First we introduce a result that will be used in this section. Under the bounded stochastic gradient variance assumption (Assumption 4), we have that conditioned on $w^t$,

$$\mathbb{E}_t \left[ \|g^t\|^2 \right] \leq \frac{\tau^2}{b} + \|\nabla F(w^t)\|^2, \tag{60}$$

where $b$ refers to the mini-batch size to obtain gradient $g^t$, i.e., $g^t \leftarrow \frac{1}{b} \sum_{i \in B} g^{i,t}$. This lemma is proved in Zaheer et al. (2018). The per-coordinate version of this result is that for $j \in [d]$,

$$\mathbb{E}_t \left[ (g_j^t)^2 \right] \leq \frac{\tau_j^2}{b} + \left( \nabla_j F(w^t) \right)^2, \tag{61}$$

and $\sum_{j \in [d]} \tau_j^2 = \tau^2$.

As we assume $F(w)$ is $L$-smooth, at each iteration $t$,

$$F(w^{t+1}) \leq F(w^t) + \langle \nabla F(w^t), w^{t+1} - w^t \rangle + \frac{L}{2} \left\| w^{t+1} - w^t \right\|^2. \tag{62}$$

Based on the updating rule of Algorithm 1, we have

$$F(w^{t+1}) \leq F(w^t) + \langle \nabla F(w^t), w^{t+1} - w^t \rangle + \frac{L}{2} \left\| w^{t+1} - w^t \right\|^2 \tag{63}$$

$$= F(w^t) - \alpha^t \left\langle \nabla F(w^t), \frac{g^t}{D^t} + N^t \right\rangle + \frac{(\alpha^t)^2 L}{2} \left\| \frac{g^t}{D^t} + N^t \right\|^2, \tag{64}$$

where $N \in \mathbb{R}^d$ and $N_j \sim \mathcal{N} \left( 0, \frac{\sigma^2 C^2}{b^2} \right)$ with noise multiplier $\sigma$ and clipping threshold $C$, and $D^t$ satisfies that

$$D^t \leftarrow \begin{cases} \mathbf{1} & \text{if } t \bmod 2s \leq s, \\ \sqrt{v} + \epsilon & \text{otherwise.} \end{cases} \tag{65}$$

Take expectation with respect to samples at the $t$-th iteration and $N^t$,

$$\mathbb{E}_t[F(w^{t+1})] \leq F(w^t) - \alpha^t \left\langle \nabla F(w^t), \frac{\nabla F(w^t)}{D^t} \right\rangle + \frac{(\alpha^t)^2 L}{2} \mathbb{E}_t \left[ \left\| \frac{g^t}{D^t} \right\|^2 \right] + \frac{d(\alpha^t)^2 L}{2b^2} \sigma^2 C^2$$

$$= F(w^t) - \alpha^t \sum_{j \in [d]} \frac{(\nabla_j F(w^t))^2}{D_j^t} + \frac{(\alpha^t)^2 L}{2} \sum_{j \in [d]} \frac{\mathbb{E}_t \left[ (g_j^t)^2 \right]}{(D_j^t)^2} + \frac{d(\alpha^t)^2 L}{2b^2} \sigma^2 C^2, \tag{66}$$

where we have used the fact that $N^t$ is a zero-mean random variable independent of $w^t$, and $D^t$ is independent of samples at time $t$. We need to consider two cases.

1. **DP-SGD at the $t$-th iteration**
   In this case, $D^t = 1$. Hence plugging in

$$\mathbb{E}_t\left[(g_j^t)^2\right] \leq \frac{\tau_j^2}{b} + \left(\nabla_j F(w^t)\right)^2, \tag{67}$$

we have

$$\mathbb{E}_t\left[F(w^{t+1})\right] \leq F(w^t) - \left(\alpha^t - \frac{(\alpha^t)^2 L}{2}\right)\|\nabla F(w^t)\|^2 + (\alpha^t)^2 L\left(\frac{\tau^2}{2b} + \frac{\sigma^2 C^2 d}{2b^2}\right). \tag{68}$$

Under constant learning rate, let $\alpha^t = \alpha \leq \frac{1}{L}$,

$$\mathbb{E}_t\left[F(w^{t+1})\right] \leq F(w^t) - \frac{\alpha}{2}\|\nabla F(w^t)\|^2 + (\alpha^t)^2 L\left(\frac{\tau^2}{2b} + \frac{\sigma^2 C^2 d}{2b^2}\right). \tag{69}$$

Taking expectation on both sides gives

$$\frac{\alpha}{2}\mathbb{E}\left[\|\nabla F(w^t)\|_2^2\right] \leq \mathbb{E}[F(w^t)] - \mathbb{E}[F(w^{t+1})] + (\alpha^t)^2 L\left(\frac{\tau^2}{2b} + \frac{\sigma^2 C^2 d}{2b^2}\right). \tag{70}$$

2. **Private RMSProp at the $t$-th iteration**
   We have

$$\mathbb{E}_t[F(w^{t+1})] \leq F(w^t) - \alpha^t \sum_{j\in[d]} \frac{[\nabla F(w^t)]_j^2}{\sqrt{v_j^t} + \epsilon} + \frac{(\alpha^t)^2 L}{2\epsilon} \sum_{j\in[d]} \frac{\mathbb{E}_t[(g_j^t)^2]}{\sqrt{v_j^t} + \epsilon^t} + \frac{d(\alpha^t)^2 L\sigma^2 C^2}{2b^2}. \tag{71}$$

Plugging in $\mathbb{E}_t\left[(g_j^t)^2\right] \leq \frac{\tau_j^2}{b} + \left(\nabla_j F(w^t)\right)^2$ results in

$$\mathbb{E}_t[F(w^{t+1})] \tag{72}$$

$$\leq F(w^t) - \alpha^t \sum_{j\in[d]} \frac{[\nabla F(w^t)]_j^2}{\sqrt{v_j^t} + \epsilon} + \frac{(\alpha^t)^2 L}{2\epsilon} \sum_{j\in[d]} \frac{\sigma_j^2}{\left(\sqrt{v_j^t} + \epsilon\right)b}$$

$$+ \frac{(\alpha^t)^2 L}{2\epsilon} \sum_{j\in[d]} \frac{[\nabla F(w^t)]_j^2}{\sqrt{v_j^t} + \epsilon} + \frac{d(\alpha^t)^2 L\sigma^2 C^2}{2b^2} \tag{73}$$

$$= F(w^t) - \left(\alpha^t - \frac{(\alpha^t)^2 L}{2\epsilon}\right) \sum_{j\in[d]} \frac{[\nabla F(w^t)]_j^2}{\sqrt{v_j^t} + \epsilon} + \frac{(\alpha^t)^2 L}{2\epsilon} \sum_{j\in[d]} \frac{\tau_j^2}{\left(\sqrt{v_j^t} + \epsilon\right)b} + \frac{d(\alpha^t)^2 L\sigma^2 C^2}{2b^2} \tag{74}$$

$$\leq F(w^t) - \left(\alpha^t - \frac{(\alpha^t)^2 L}{2\epsilon}\right) \sum_{j\in[d]} \frac{[\nabla F(w^t)]_j^2}{\sqrt{v_j^t} + \epsilon} + (\alpha^t)^2 L\left(\frac{\tau^2}{2\epsilon^2 b} + \frac{d\sigma^2 C^2}{2b^2}\right). \tag{75}$$

Taking expectation on both sides yields

$$\mathbb{E}[F(w^{t+1})] \leq \mathbb{E}[F(w^t)] - \left(\alpha^t - \frac{(\alpha^t)^2 L}{2\epsilon}\right) \sum_{j\in[d]} \mathbb{E}\left[\frac{[\nabla F(w^t)]_j^2}{\sqrt{v_j^t} + \epsilon}\right] + (\alpha^t)^2 L\left(\frac{\tau^2}{2\epsilon^2 b} + \frac{d\sigma^2 C^2}{2b^2}\right). \tag{76}$$

We need to lower bound $\sum_{j\in[d]} \mathbb{E}\left[\frac{[\nabla F(w^t)]_j^2}{\sqrt{v_j^t} + \epsilon}\right]$. We know from Holder's inequality that $\mathbb{E}[\langle u, v\rangle] \leq \mathbb{E}[\|u\|_1]\mathbb{E}[\|v\|_\infty]$. Now note that

$$\mathbb{E}\left[\|\nabla F(w^t)\|^2\right] = \mathbb{E}\left[\left\langle \frac{|\nabla F(w^t)|^2}{D^t}, D^t\right\rangle\right] \leq \mathbb{E}\left[\left\|\frac{(\nabla F(w^t))^2}{D^t}\right\|_1\right] \mathbb{E}\left[\|D^t\|_\infty\right] \tag{77}$$

$$\leq \mathbb{E}\left[\left\|\frac{(\nabla F(w^t))^2}{D^t}\right\|_1\right] (\sqrt{M} + \epsilon). \tag{78}$$

Hence

$$\sum_{j \in [d]} \mathbb{E}\left[\frac{(\nabla_j F(w^t))^2}{D_j^t}\right] \geq \frac{\mathbb{E}[\|\nabla F(w^t)\|^2]}{\sqrt{M} + \epsilon} \tag{79}$$

and

$$\mathbb{E}[F(w^{t+1})] \leq \mathbb{E}[F(w^t)] - \left(\alpha^t - \frac{(\alpha^t)^2 L}{2\epsilon}\right)\frac{\mathbb{E}\left[\|\nabla F(w^t)\|^2\right]}{\sqrt{M} + \epsilon} + (\alpha^t)^2 L \left(\frac{\tau^2}{2\epsilon^2 b} + \frac{d\sigma^2 C^2}{2b^2}\right). \tag{80}$$

Let $\alpha^t = \alpha \leq \frac{\epsilon}{L}$, we obtain

$$\mathbb{E}[F(w^{t+1})] \leq \mathbb{E}[F(w^t)] - \frac{\alpha}{2(\sqrt{M} + \epsilon)}\mathbb{E}\left[\|\nabla F(w^t)\|^2\right] + (\alpha^t)^2 L \left(\frac{\tau^2}{2\epsilon^2 b} + \frac{d\sigma^2 C^2}{2b^2}\right). \tag{81}$$

Combining the two cases, for any $t$, we have

$$\mathbb{E}[\|\nabla F(w^t)\|^2] \tag{82}$$

$$\leq \frac{2(\sqrt{M} + 1)}{\alpha}\left(\mathbb{E}[F(w^t)] - \mathbb{E}[F(w^{t+1})]\right) + 2\alpha L(\sqrt{M} + 1)\left(\frac{\tau^2}{2\epsilon^2 b} + \frac{d\sigma^2 C^2}{2b^2}\right). \tag{83}$$

Taking a telescope sum results in

$$\frac{1}{T}\sum_{t=1}^T \mathbb{E}[\|\nabla F(w^t)\|^2] \leq \frac{2(\sqrt{M} + 1)F(w^1)}{\alpha T} + 2\alpha L(\sqrt{M} + 1)\left(\frac{\tau^2}{2\epsilon^2 b} + \frac{d\sigma^2 C^2}{2b^2}\right), \tag{84}$$

where $M := C^2 + \frac{\sigma^2 C^2}{sb^2}$.

## C  EXPERIMENTAL DETAILS AND ADDITIONAL RESULTS

### C.1  DATASETS

**IMDB** (Maas et al., 2011) is a binary classification dataset on sentiment analysis for movie reviews that includes 25,000/25,000 training/test samples. Each sample is a review under a vocabulary size of 10,000. We train a logistic regression model with 10,001 parameters.

**StackOverflow** (Kaggle, 2022; TensorFlow Federated, 2022) is a large-scale text dataset containing questions and answers from Stack Overflow. We focus on the task of classifying the tag(s) of a given sentence described in TensorFlow Federated (2022), though we focus on the usual centralized training setting instead of a federated setting. We randomly sample 246,092 sentences for training and 61,719 for testing, where each sentence is described by 10,000 features. We format the task as a 500-class classification problem, and the resulting model has roughly 5 million parameters.

**MovieLens-100k** (Harper & Konstan, 2015) is a movie review dataset commonly used for recommendation systems. It contains 100,000 movie ratings from 943 users on 1,682 items ($\approx 6\%$ non-zero entries). We study a (non-convex) matrix factorization task with embedding size 100, thus totaling 262,500 parameters. We treat each non-zero entry as a 'record' for differential privacy, and randomly partition them for training and evaluation.

### C.2  HYPERPARAMETERS

Unless otherwise stated, we fix the following hyperparameters in our experiments: for IMDB, StackOverflow, and MovieLens respectively, we train for 100/50/50 epochs with batch size 64 and privacy $\delta = 10^{-5}/10^{-6}/10^{-6}$. We then perform a grid search on other hyperparameters:

- *Learning rates*: We grid search over $\{0.03, 0.1, 0.3, 1, 3, 5\}$ for SGD / AdaGrad update rules and from $\{0.001, 0.003, 0.01, 0.03, 0.1, 0.3, 1, 3\}$ for the RMSProp update rule.

- *Per-example clipping thresholds*: We grid search over $\{0.1, 0.25, 0.5, 1\}$ when performing per-example clipping on clean gradients *without preconditioning* (e.g. for DP-SGD updates), and over $\{0.1, 0.25, 0.5, 1, 2, 3, 5\}$ when clipping *preconditioned* clean gradients (e.g. for $\mathrm{DP}^2$ updates in adaptive iterations). The rationale is that, in general, the preconditioned gradient norms are usually larger than those without preconditioning (recall from Section 3.2 that we apply preconditioning *before* privatization in $\mathrm{DP}^2$). For AdaDPS and $\mathrm{DP}^2$-RMSProp, we also tried a few values of even larger clip thresholds ($\geq 10$) though we did not perform a full sweep for other hyperparameters at those values due to computational constraints.

- *Delay parameter $s$*: For all datasets, $s$ (i.e., the number of optimization steps) is chosen heuristically as a function of the number of steps in an epoch. When reporting the best results (e.g. Figure 4, Figure 5), we search over $s \in \{195, 390, 780\}$ (roughly 0.5, 1, 2 epochs respectively) for IMDB (390 steps/epoch); $s \in \{100, 300, 1000, 3000\}$ for StackOverflow (3845 steps/epoch); and $s \in \{1250, 15625, 31250, 50000\}$ for MovieLens (1250 steps/epoch).

- *Adaptivity $\epsilon$*: In our settings, the adaptivity parameter $\epsilon$ for RMSProp/AdaGrad (in the denominator $D^t = \sqrt{v} + \epsilon$) would affect the amount of adaptivity as well as the norms of preconditioned gradients, which may in turn influence the privacy-utility trade-off under per-example clipping. We tune $\epsilon$ over a small grid of $\{10^{-2}, 10^{-3}, 10^{-5}, 10^{-7}\}$.

All reported results use the best hyperparameter configurations, which are selected using training set metrics (as overfitting generally does not occur under DP noise). To facilitate reproducibility, we summarize the tuned hyperparameters for the main experiments and the ablation studies in Table 2 and Table 3 below respectively.

| Dataset | DP-SGD | DP-RMSProp | PDA-DPMD | AdaDPS (w/ RMSProp) | DP$^2$-RMSProp |
|---|---|---|---|---|---|
| IMDB | (**5**, 0.5) | (0.3, **0.1**, 10$^{-3}$) | (**5**, 0.5) | (1, **5**, 10$^{-3}$) | (0.1, **3**, 0.5, **5**, **10$^{-7}$**, **195**) |
| StackOverflow | (3, 0.25) | (0.03, **0.1**, 10$^{-3}$) | (3, 0.25) | (0.4, **5**, 10$^{-3}$) | (0.3, 0.3, 0.25, **5**, 10$^{-5}$, 1000) |
| MovieLens | (0.1, **1**) | (**0.001**, 0.5, 10$^{-3}$) | (0.1, **1**) | (0.01, **10**, **10$^{-2}$**) | (0.1, 0.03, **1**, **5**, 10$^{-3}$, 31250) |

Table 2: **Tuned hyperparameters for different methods across three datasets**. For DP-SGD and PDA-DPMD, the values refer to (LR, clip); for DP-RMSProp and AdaDPS, the values refer to (LR, clip, adaptivity $\epsilon$); and for DP$^2$, the values refer to (LR for SGD iters, LR for RMSProp iters, clip for SGD iters, clip for RMSProp iters, adaptivity $\epsilon$, delay $s$). **Bold values** were experimented on the edges of the hyperparameter grids.

| Dataset | Ablation Variant1 | Ablation Variant 2 |
|---|---|---|
| IMDB | (3.0, 0.1, 0.5, 2.0, **10$^{-7}$**, **780**) | (0.3, 0.3, 0.25, 10$^{-3}$, **780**) |
| StackOverflow | (1.0, 1.0, 1.0, 1.0, 10$^{-5}$, **1000**) | (0.3, **0.001**, 0.25, 10$^{-5}$, **1000**) |

Table 3: **Tuned hyperparameters for ablation studies (Section 5.3) on IMDB and StackOverflow**. Both variants use the RMSProp update rule for the adaptive steps. **Bold values** were experimented on the edges of the hyperparameter grids. For Variant 1 and 2 respectively, the values refer to (LR for SGD iters, LR for RMSProp iters, clip for SGD iters, clip for RMSProp iters, adaptivity $\epsilon$, delay $s$) and (LR for SGD iters, LR for RMSProp iters, clip for both SGD/RMSProp iters, adaptivity $\epsilon$, delay $s$). Note that for Variant 2 the clipping threshold do not need to be tuned separately for SGD/RMSProp iters as it applies to preconditioned gradients in both cases.

## C.3 RESULTS FOR DP$^2$-ADAGRAD

The DP$^2$ framework can be applied to a range of adaptive methods beyond RMSProp mostly discussed in the main text. We extend DP$^2$ to the AdaGrad update rule (with only one line of code change, see Section D), and benchmark its convergence and privacy-utility trade-offs. In Figure 8 and Figure 9, the results indicate that DP$^2$-AdaGrad, like DP$^2$-RMSProp, can consistently and substantially improve over the baselines in terms of both convergence and absolution performance, demonstrating the generality of DP$^2$ to other adaptive optimizers.

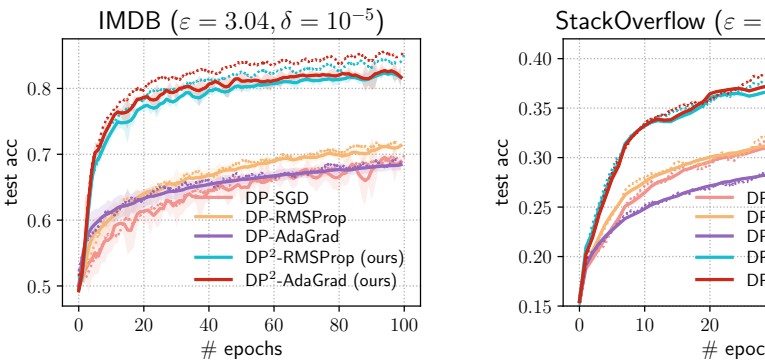

Figure 8: (**Extension of Figure 4 to the AdaGrad update rule**) Test accuracy of DP$^2$ compared to DP-SGD, DP-RMSProp, and DP-AdaGrad on IMDB and StackOverflow. Dotted lines denote training performance.

## C.4 EFFECTS OF INCREASING COMPUTATIONAL BUDGETS

When differential privacy introduces a large utility gap between private and non-private training, one approach to improving the privacy-utility trade-off is to increase computational costs by using larger batch sizes under fixed numbers of steps. The noise multiplier needs to increase to achieve the same privacy target, while the overall privacy noise may still be reduced due to the larger batch size. This technique may be adopted in practice when we want to prioritize the utility of private optimization

under fixed privacy budgets. In Figure 9 (right), we explore the effect of such increased computation on StackOverflow. With a $4\times$ factor increase in computational cost ($4\times$ larger batch sizes with the same number of training iterations), we observe that the privacy/utility trade-off of all methods can be substantially improved, narrowing the utility gap to non-private training. In particular, observe that the absolute performance improvement of $\mathtt{DP}^2$ over the vanilla DP baselines remains similar.

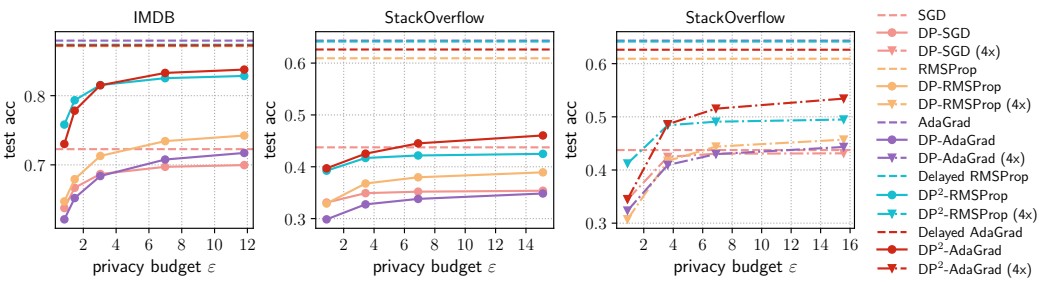

Figure 9: (**Extension of Figure 5 to the AdaGrad update rule and increased computational cost**) Privacy/utility trade-offs of $\mathtt{DP}^2$ compared to DP-SGD, DP-RMSProp, and DP-AdaGrad on IMDB and StackOverflow. "($4\times$)" denotes increasing the batch size and the number of epochs simultaneously by a factor of 4 and picking the appropriate noise multiplier to arrive at similar privacy costs ($\varepsilon$).

## C.5 ADDITIONAL RESULTS FOR ABLATION STUDIES

Table 4 summarizes the results for ablation studies on IMDB, StackOverflow, and MovieLens, and Figure 10 reports test accuracies on IMDB and StackOverflow during optimization. The variants are discussed in Section 5.3 and complete algorithms are presented in Appendix D. We observe that $\mathtt{DP}^2$ indeed consistently outperforms the two (weaker) variants on all datasets, thus verifying our design choices for $\mathtt{DP}^2$. In particular, note that the utility drop of variant 2 (adding noise before preconditioning) on StackOverflow is more significant compared to that on IMDB; we argue that this is due to StackOverflow being a high-dimensional learning task (roughly 5 million model parameters) and thus the detrimental effect of preconditioning per-coordinate noise is larger.

| Dataset | Variant1 | Variant 2 | $\mathtt{DP}^2$-RMSProp |
|---|---|---|---|
| IMDB $\uparrow$ | $.799 \pm .006$ | $.643 \pm .007$ | $\mathbf{.815} \pm .011$ |
| StackOverflow $\uparrow$ | $.382 \pm .002$ | $.265 \pm .004$ | $\mathbf{.391} \pm .001$ |
| MovieLens $\downarrow$ | $3.32 \pm .088$ | $3.18 \pm .066$ | $\mathbf{2.78} \pm .054$ |

Table 4: Summary of ablation studies on all three datasets.

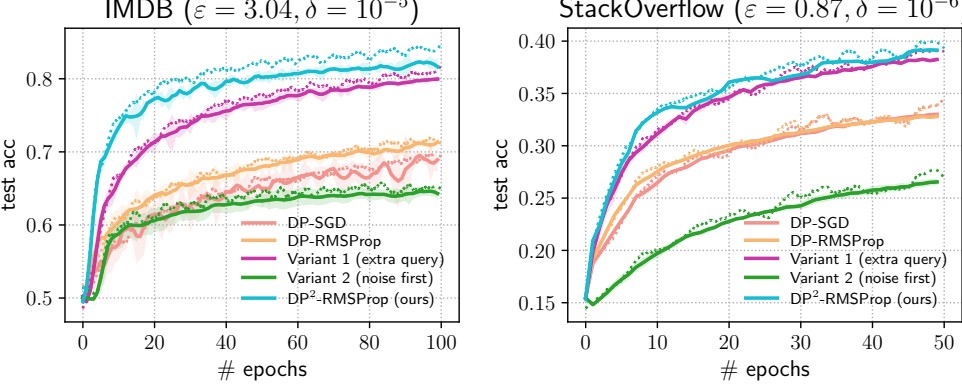

Figure 10: Test accuracies for ablation studies on $\mathtt{DP}^2$. Dotted lines correspond to training metrics.

### C.6 ADDITIONAL RESULTS FOR COMPARISON WITH PUBLIC DATA-ASSISTED METHODS

Figure 11 extends the results in Section 5.2 with convergence plots on IMDB and StackOverflow. On IMDB, we observe that despite not using any auxiliary information, the convergence of $\text{DP}^2$-RMSProp is comparable with that of AdaDPS-RMSProp (Li et al., 2022) which uses 1% of training data as the public data (250 examples) to approximate the preconditioner. On StackOverflow where the same public split of 1% corresponds to 2460 examples, we observe that AdaDPS-RMSProp can outperform $\text{DP}^2$. On the other hand, the extra public data do not help PDA-DPMD outperform $\text{DP}^2$.

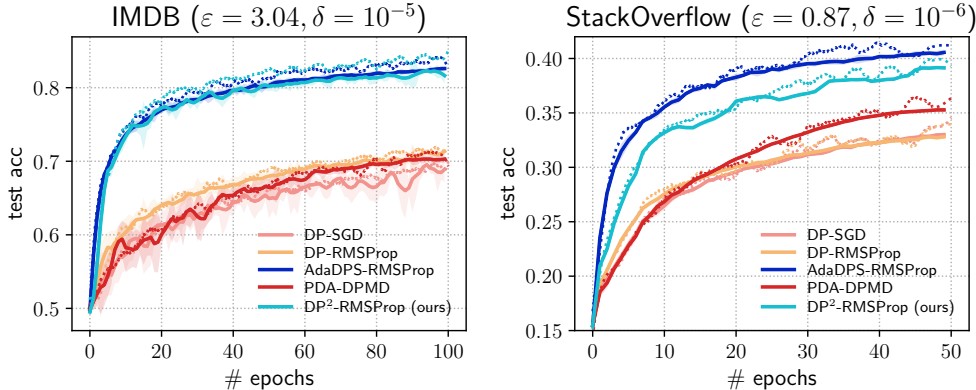

Figure 11: Test accuracies of $\text{DP}^2$ compared against recent private (adaptive) methods that leverage public data (Amid et al., 2022; Li et al., 2022). Dotted lines correspond to training metrics.

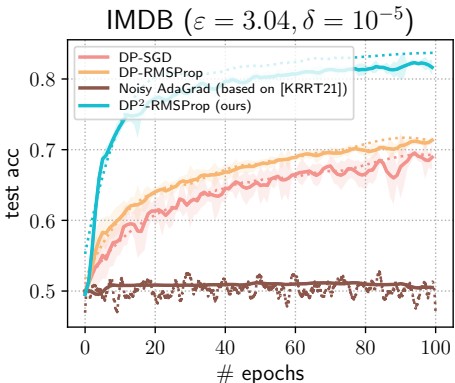

Figure 12: Comparing $\text{DP}^2$ against a noisy AdaGrad variant based on Kairouz et al. (2021a) where the gradients and the preconditioner are privatized separately.

In Figure 12, we additionally implement a private AdaGrad method proposed in Kairouz et al. (2021a) that also leverages public data. Specifically, in each iteration, the algorithm clips and adds independent noise to both the clean gradients and the preconditioner estimated using clean gradients; it then uses public data to estimate a gradient subspace onto which to project the clipped/noised preconditioner in order to reduce the effect of noise; finally, it preconditions the noisy gradient with the noisy preconditioner and takes an update step. Our implementation differs from Kairouz et al. (2021a) in that we use the diagonal form of the preconditioner instead of the full matrix form. To estimate the gradient subspace, we follow the approach described in Zhou et al. (2021) where the projection matrix $V \in \mathbb{R}^{d \times k}$ where $d$ is the number of parameters and $k$ is the dimension of the subspace is obtained by taking the top-$k$ eigenspace of $M^t$ with

$$M^t = \frac{1}{|X_{\text{pub}}|} \sum_{x^i \in X_{\text{pub}}} \nabla_{w^t} f\left(x^i; w^t\right) \nabla_{w^t} f\left(x^i; w^t\right)^\top$$

where $X_{\text{pub}}$ is the set of public examples. Unfortunately, we have not obtained a satisfactory result for this noisy AdaGrad algorithm. We remark that since the method is extremely computationally

expensive (involves computing the eigendecomposition of a $d \times d$ matrix with $d = 10001$ at every iteration), further hyperparameter tuning may help improve the performance. However, our ablation studies (Section 5.3 and Appendix C.5) may shed light on the current observations since this method privatizes gradients before preconditioning.

## D    ALGORITHMS

For completeness, we present all algorithms mentioned in the main text in detail.

- **Non-private version of** `DP`$^2$: only changing Line 9 in Algorithm 1 to

$$\tilde{g}^t \leftarrow \frac{1}{b} \sum_{i \in B} \frac{g^{i,t}}{D^t}$$

- `DP`$^2$ **with the AdaGrad update rule (**`DP`$^2$**-AdaGrad)**: only changing Line 5 in Algorithm 1 to

$$v \leftarrow v + \left( G^t / s_1 \right)^2$$

- `DP`$^2$ **with Yogi's additive update rule (**`DP`$^2$**-Yogi)**: only changing Line 5 in Algorithm 1 to

$$v \leftarrow v + (1 - \beta)\text{sign}(G^t / s_1 - v^2) \left( G^t / s_1 \right)^2$$

- **Ablation variant 1 (extra query) with delayed preconditioners**: see Algorithm 2. Observe that the clean batch gradients $\{g^{i,t}\}_{i \in B}$ get privatized twice in most iterations (when $(t-1) \bmod s \neq 0$), increasing the total privacy cost.

- **Ablation variant 2 (noise before preconditioning) with delayed preconditioners**: in Line 9 of Figure 1, privatize the batch gradients with the following replacement:

$$\tilde{g}^t \leftarrow \frac{1}{b} \left( \sum_{i \in B} \text{clip} \left( g^{i,t}, C \right) + \mathcal{N} \left( \mathbf{0}, \sigma^2 C^2 \right) \right) / D^t$$

---

**Algorithm 2:** Ablation variant 1 (extra query) using delayed preconditioners

---

**Input:** $T$, batch size $b$, noise multiplier $\sigma$, clipping thresholds $C_1$, $C_2$, initial model $w^0 \in \mathbb{R}^d$, $v = \mathbf{0}$, constant $\epsilon \in \mathbb{R}_+$, learning rate schedule $\alpha^t$, moving average parameters $\beta$, delay steps $s$

1   Set accumulator $G^0 \leftarrow \mathbf{0}$

2   **for** $t = 1, \cdots, T$ **do**

3     Uniformly randomly sample a mini-batch $B$ with size $b$ from private training data

4     Get individual gradients for sample $i \in B$: $g^{i,t} \leftarrow \nabla f(x^i; w^{t-1})$

5     Privatize the gradients using the Gaussian mechanism:

$$\tilde{g}^t \leftarrow \frac{1}{b} \left( \sum_{i \in B} \text{clip}\left( g^{i,t}, C_1 \right) + \mathcal{N}\left( \mathbf{0}, \sigma^2 C_1^2 \right) \right)$$

     Accumulate the private gradients $\tilde{g}^t$: $G^t \leftarrow G^{t-1} + \tilde{g}^t$

6     **if** $(t-1) \bmod s = 0$ **then**

7       Update moment estimates: $v \leftarrow \beta v + (1 - \beta)\left( G^t / s \right)^2$

8       Reset accumulator: $G^t \leftarrow \mathbf{0}$

9       **Set final gradient**: $\bar{g}^t \leftarrow \tilde{g}^t$

10     **else**

11       Privatize the **clean, preconditioned** gradients using the Gaussian mechanism:

$$\hat{g}^t \leftarrow \frac{1}{b} \left( \sum_{i \in B} \text{clip}\left( \frac{g^{i,t}}{\sqrt{v} + \epsilon}, C_2 \right) + \mathcal{N}\left( \mathbf{0}, \sigma^2 C_2^2 \right) \right)$$

      **Set final gradient**: $\bar{g}^t \leftarrow \hat{g}^t$

12     Update model parameters $w$:

$$w^t \leftarrow w^{t-1} - \alpha^t \bar{g}^t$$

13   **return** $w^T$

---

