# OpenReview forum: "Differentially Private Adaptive Optimization with Delayed Preconditioners"
_ICLR.cc/2023/Conference — ICLR 2023 poster_

### Official Review · Reviewer_P2DR · 2022-10-25

**Confidence:** 3
**Clarity, Quality, Novelty And Reproducibility:** This paper has minor technical flaws.…
**Correctness:** 3
**Technical Novelty And Significance:** 2
**Empirical Novelty And Significance:** 3
**Recommendation:** 3

**Strength And Weaknesses:**

Strength:
This paper is easy to follow, and the target problem is interesting.

Weakness:
1. Assumption 1 is quite wired. For any iteration t, this should be proved theoretical rather than simply assuming it holds for all $w^{t}$.
2. What is the definition of $\|\|\cdot\|\|_{D^{t}}$?
3. It has been discussed about the trade-offs between delay and noise, which suggest there exists an optimal $\nu$. Could the authors investigate the trade-offs numerically?

**Summary Of The Paper:**

This paper provided a differential private adaptive training with delayed preconditioners to avoid using auxiliary data for private optimization. Some theoretical results and several numerical studies are explored to demonstrate the effectiveness of the proposed method.

**Summary Of The Review:**

This paper simply added delayed preconditioners in existing differential private algorithms. The contribution seems to be incremental.

---

> ### Author Response · Authors · 2022-11-15
> **Response**
>
> We thank the reviewer for their time and review of the paper.
>
> **[Assumption 1 (Bounded Domain Assumption)]** The bounded domain assumption (for all iterations) is commonly used in adaptive optimization literature (e.g., Levy et al., 2018, Reddi et al., 2018, Asi et al., 2021, Li et al., 2022). This can also be easily achieved by adding an additional projection step at each iteration (which does not affect our convergence analysis due to the contraction property of projections).
>
> **[Definition of $||\cdot||_{D^t}$]**  We note $||x||_A$ denotes $\sqrt{x^T A x} $ for a PSD matrix $A$. With abuse of notations, we also use $||x||_v := \sqrt{x^T \text{diag}(v) x } $ when $v$ is a vector with non-negative entries, i.e., expanding $v$ to be a diagonal PSD matrix. Please also see the ‘Notation’ paragraph at the end of Section 2 in our original submission.
>
> **[Trade-offs Between Delay and Noise]** Thanks for the question. In our original submission, we have already investigated the trade-offs between delay and noise both analytically (Section 4.1) and numerically (Section 5.2, Figure 6). To address the reviewer’s question, we explicitly demonstrate the effects of the delay parameter $s$ in Figure 6. We find that (a) there exist specific values of $s$ that achieve the greatest improvements, and (b) nearly all instantiations of DP$^2$ improve upon the baselines.
>
> **[Contributions]** Finally, we aim to respond to the reviewer’s assessment of novelty: We believe both our problem and our solution are entirely novel. Despite its simplicity, the general idea of using delayed preconditioners to improve private adaptive optimization has not been studied previously (as we discuss, even in non-private settings, this technique has only been briefly mentioned as a heuristic in one prior work that we are aware of, for different purposes). The exact method we propose for private optimization is also novel, and is not as straightforward as just ‘adding delayed preconditioners to existing methods’. It was in fact quite unclear how to construct and properly apply delayed preconditioners in conjunction with differential privacy---we discovered this by exploring several possible variants (see Section 5.4, Figure 7). Overall, our work provides one of the first principled yet highly effective solutions we are aware of for performing private adaptive optimization without access to public data. We rigorously analyze DP$^2$ in both convex and non-convex settings, and conduct extensive experiments to showcase the effectiveness of DP$^2$ across a range of benchmarks.
>
> Please let us know if you have additional questions or concerns about our submission.
>
> [Levy et al., 2018] Online adaptive methods, universality and acceleration \
> [Reddi et al., 2018] On the convergence of Adam and beyond \
> [Asi et al., 2021] Private adaptive gradient methods for convex optimization \
> [Li et al., 2022] Private adaptive optimization with side information

---

### Official Review · Reviewer_vZ2E · 2022-10-26

**Confidence:** 3
**Correctness:** 3
**Technical Novelty And Significance:** 3
**Empirical Novelty And Significance:** 3
**Recommendation:** 8

**Clarity, Quality, Novelty And Reproducibility:**


As said the paper is very well written. Here few additional comments:

- How big were the model sizes used in the experiments? Would this have some effect?

- Minor: to me it seems that $G^t$ is a dummy variable between from iterations $s_1$ to $s_1 + s_2$ (mod $s_1 + s_2$). This is cosmetics but perhaps that could be somehow written differently.

- It was for me a bit difficult to grasp how exactly do we benefit from the staleness of the gradients. I.e. we update the accumulator for $s_1$ steps and preconditioner after eevry $s_1 + s_2$ steps. Wouldn't the weighted summing averaging anyhow cancel the effect of DP noise? is there something more at play, that it is actually beneficial to use slightly stale preconditioners?



**Strength And Weaknesses:**


Pros:

- This seems like a novel idea and all these ideas seem like a natural thing to do for DP-SGD (averaging, using preconditioning before clipping)
- Very strong experimental results for the proposed method
- The paper is very well written and the convergence analysis seems convincing. For example, for convex problems, it shown that smaller amount of noise is added compared with DP-SGD which leads to faster convergence.

Cons:

- Slight deficit of the convergence analysis: the analysis does not seem to improve over the state-of-the-art results for adaptive DP optimizers for non-convex problems, although the same asymptotics are obtained (as far as I see). This means that the strong experimental results are not reflected in the non-convex analysis.
- I was expecting also some standard DP-SGD comparisons (MNIST, CIFAR10) etc. Could you comment, would the behaviour for this method be then similar?



**Summary Of The Paper:**

Background: DP-SGD gives guarantees for the whole history of gradients. It very difficult to obtain guarantees that would hold only for the final model, so then it is natural to think, how could we use the history of gradients since due to post-processing that will not have any additional privacy cost.

The paper proposes a novel RMSprop type of preconditioner for DP-SGD. There are several key steps to reduce the effect of DP noise in the preconditioner. Commonly the preconditioners are plain non-DP SGD preconditioners applied to the noisy gradients, however this work tailors the RMSprop approach to DP-SGD.

Important steps include

-average the preconditioner over noisy gradients. Preconditioner here means vector v with which the gradients are scaled element-wise before clipping. Averaging over the history of gradients reduces the effect of DP noise.
-carry out the preconditioning before clipping, instead of afterwards (when e.g. using naively preconditioners to DP-SGD)
-update the preconditioner only once in a while (for a short window of iterations update the accumulator with which the preconditioned is updated), then carry out preconditioning with the old one.

**Summary Of The Review:**

All in all, I think the results (especially experimental) seem very convincing and the ideas natural, so I recommend acceptance.

---

> ### Author Response · Authors · 2022-11-15
> **Response**
>
> We greatly appreciate the reviewer’s insightful and valuable review.
>
> **[Non-Convex Results]** The reviewer is correct that we don’t achieve constant improvements in our non-convex rates, despite seeing improvements in our experiments. Therefore the non-convex rates cannot demonstrate the tradeoffs between delay and privacy noise. Without any delay, proving the benefits of adaptivity for non-convex problems in general remains an open problem even in non-private learning (see shared response), which we leave as a direction of future work.
>
> **[Results on MNIST/CIFAR-10]** Thanks for the detailed question. We tested DP$^2$ and baseline methods on MNIST, and observed that all methods performed very similarly. Indeed, while MNIST and CIFAR-10 may be commonly used in the privacy literature, adaptive optimizers typically do not have a significant advantage over SGD on those datasets (in private or in non-private settings). Our empirical study instead focused on datasets where adaptivity is crucial, and aimed to understand whether our private optimization method can retain improvements in those settings. We will make it more clear that DP$^2$ is especially useful for settings where adaptive optimizers tend to excel (e.g, tasks with sparse gradients).
>
> **[Other Clarification]** Thanks for the additional comments. (a) The model size ranges from 10k to 5 million (as reported in Section 5.1). Empirically, we observe consistent improvements across all models. (b) Yes, $G^t$ can be seen as a dummy variable in that it is not used for certain iterations. (c) Averaging over noisy gradients would result in a less noisy but delayed preconditioner. It is correct that the independent noise across iterations will be canceled out, which is precisely the reason for the improved performance of DP$^2$. The staleness of preconditioners is not beneficial per se, but our motivation is that such staleness does not hurt either (Section 3.1).

---

### Official Review · Reviewer_hof5 · 2022-10-28

**Confidence:** 3
**Correctness:** 3
**Technical Novelty And Significance:** 2
**Empirical Novelty And Significance:** 3
**Recommendation:** 6

**Clarity, Quality, Novelty And Reproducibility:**

This paper is easy to follow. The quality and significance can be improved. The experiments could be reproducible with the details provided in this paper.


**Strength And Weaknesses:**

Strength:
1. The proposed approach is well-motivated. It is reasonable to use a delayed preconditioner based on the empirical observation that the preconditioner value distribution is consistent between successive epochs.
2. The experiment is comprehensive. This paper has experimented on multiple datasets, comparing the proposed algorithm with existing private optimizers and very recent private algorithms with public data. This paper also empirically studies privacy/utility trade-offs, and delay parameters.
3. The writing is good. This paper is easy to follow.


Weakness:
1. The theoretical convergence bound for convex optimization is very hard to interpret. It is unclear to understand the rate of convergence in terms of dependence on iteration time 'T', delay parameter 's' or 'v', and dimension dependence 'd'. It will be good to clarify those dependencies in the bound and compare Theorem 2 with existing bounds which can make the contribution of this paper clear.
2. There is inconsistency regarding the dependence on noise variance in Theorem 2 and Theorem 3. In the convex case (Theorem 2), the noise variance is scaled by the preconditioner, i.e., \|N^t \|^2_{D^t}, which shows the noise is projected by the preconditioner. But in the non-convex case (Theorem 3), the noise variance is d\sigma^2 as appeared in most existing bounds. In the algorithm, the isotropic noise is added after applying the preconditioner to the gradient, which makes it more sense to have d\sigma^2 in the bound. It will be great if the authors can add some discussions about this.

**Summary Of The Paper:**

This paper proposes a new private gradient adaptive descent algorithm for convex and non-convex optimization. The proposed algorithm does a delayed preconditioner update and leverages an average noisy gradient to reduce noise in the preconditioner. The experiments show the proposed algorithm outperforms previous private gradient-based optimization algorithms.

**Summary Of The Review:**

Overall, this paper is well written. It proposes an interesting algorithm and has a comprehensive empirical study. But the paper can be improved by clarifying the contribution, improving the theoretical bound presentation, and discussing the theoretical results. I recommend  6: marginally above the acceptance threshold.

---

> ### Author Response · Authors · 2022-11-15
> **Response**
>
> We thank the reviewer for the positive evaluation of our work.
>
> **[Compare Theorem 2 with Previous Results]** Thanks for your suggestion to improve the positioning and clarity of Theorem 2. Please see ‘response to all reviewers’ for detailed comparisons between Theorem 2 and existing bounds. In summary, the dependence on $T$ is $O\left(\frac{1}{\sqrt{T}}\right)$. The exact dependence on the delay parameter $s$ is complicated and hard to abstract (which motivated us to present full results in terms of all parameters). We provide some detailed analysis in simple settings to showcase the dependence of $s$ after Theorem 2 and in Appendix A.2.
>
> **[Differences Between Convex and Non-Convex Results]** Thank you for the question. Note that isotropic noise is added to the preconditioned gradients, not the original gradient, so that we can prove the benefits of adapting to gradient geometry, as in most adaptive methods. When we use fresh preconditioners that don’t contain any privacy noise, we expect the term in the convex bounds to still be $\frac{1}{T} \sum_t \frac{1}{\sqrt{t}} \mathbb{E}[||N^t||^2_{D^t}]$ where $D^t$ is the fresh preconditioner at the $t$-th iteration. This is consistent with Li et al. (2022). Despite our empirical improvements over DP-SGD, in non-convex cases, rigorously proving the benefits of adaptivity remains an open problem even in non-private settings. Please see our shared response for more details. Thanks for the suggestion; we will add a discussion around this in the revision.
>
>
> [Li et al., 2022] Private adaptive optimization with side information

---

> > ### Comment · Reviewer_hof5 · 2022-12-07
> > **Response to the rebuttal**
> >
> > I have read the updated version and the authors' response. I appreciate that the authors have polished Theorem 2 which clearly shows the dependence in terms of T and noise variance. The authors also added a discussion comparing Theorem 2 with existing bounds. The trade-off between delay and noise is discussed after Theorem 2.
> >
> > I have a concern that the non-convex result does not reflect the trade-off between delay and noise by s. Also, it does not show the benefit of the preconditioner in the noise variance term as in the convex case. In addition, the non-convex result requires a very small step size i.e., smaller than \eps/L. Overall, I think this paper is marginally above the acceptance threshold. Thus I am keeping my score.

---

> > > ### Author Response · Authors · 2022-12-08
> > > **Author Reply**
> > >
> > > Thanks for your reply! We are glad that Theorem 2 has become more accessible. Regarding non-convex results, we would like to note that the potentially small learning rate $O(\epsilon/L)$ is common in prior works on non-convex adaptive optimization [e.g., Zaheer et al., 2018, Zhou et al., 2020]. We will add a sentence on this in the final version. Thanks again for your feedback and time reviewing our work.
> > >
> > > [Zaheer et al., 2018] Adaptive Methods for Nonconvex Optimization  \
> > > [Zhou et al., 2020] Private Stochastic Non-Convex Optimization: Adaptive Algorithms and Tighter Generalization Bounds

---

### Official Review · Reviewer_wuVW · 2022-11-01

**Confidence:** 2
**Correctness:** 3
**Technical Novelty And Significance:** 3
**Empirical Novelty And Significance:** 2
**Recommendation:** 6

**Clarity, Quality, Novelty And Reproducibility:**

There are some novel aspects to the paper.
The quality of the writing can be significantly improved.
I did not verify the reproducibility aspect.

**Strength And Weaknesses:**

The strength of the paper is the novel idea of using preconditioning in this smart way. I like that.

The main weakness of the paper is the lack of comparison with the prior work. Theorem 2 is almost impossible to interpret. Even the text following the theorem does not help the case. It would be better if the authors provide a corollary that instantiates the theorem with the parameter choice to better explain the result.

Also, the authors claim that the paper uses adaptive optimization. Does this help them improve the rate of convergence? What effect does it have on the utility guarantee? None of this is clearly spelled out. Of course, they cannot beat the lower bound due to BST14, so what is the best utility they achieve? A proper table comparing the result in this paper in context with previous works should be given. Also, what is the SCO rate? Does their algorithm also permits a nice generalization bound? Does the algorithm have to tune the staleness parameter and the temporal gradient similarity? None of these are clear in the paper.

I haven't checked the proof thoroughly due to limited time. However, the proofs I tried to verify are written in a very difficult-to-parse manner. I will reserve my final score after the discussion with the authors and when I get to read the proofs in more details.

I did not spend a lot of time going through the experiments. IMDB and StackOverflow are difficult to train datasets. The test accuracy for IMDB is really remarkable. I would really like to know more details here. It seems to be beating DP-SGD over all ranges of epsilon for all the datasets.

**Summary Of The Paper:**

The paper proposes the idea of using delayed preconditioning by an average of gradients. The idea is motivated by the supposed empirical observation that the gradient geometry does not change significantly over a period of time.

**Summary Of The Review:**

Please see above.

---

> ### Author Response · Authors · 2022-11-15
> **Response**
>
> We greatly appreciate the reviewer’s review and suggestions to improve the paper.
>
> **[Compare Theorem 2 with Previous Results]** Thanks for your suggestion to improve the readability of Theorem 2; please see ‘response to all reviewers’ for a clarification/discussion of these results.
>
> **[Benefits of Adaptivity]** You are correct that our theoretical results don’t improve upon the convergence rates of DP-SGD; this is expected as adaptive methods alone generally don’t improve rates even in non-private training. However, DP$^2$ can improve some constants, as the stochastic optimization rate is $O\left(\frac{1}{\sqrt{T}} \max_{t \in T_{\upsilon}} \mathbb{E}[||v^t+\varepsilon||\_1]\right) + O\left(\frac{1}{T} \sum_{t} \frac{1}{\sqrt{t}} \mathbb{E}[||N^t||^2\_{D^t}]\right)$, and thus we see benefits when $D^t$ is sparse. The utility bound is $O\left(\frac{1}{\sqrt{T}} \max_{t \in T_{\upsilon}} \mathbb{E}[||v^t+\varepsilon||_1]\right) + O\left(\frac{1}{\sqrt{T}} \sum_t \frac{1}{\sqrt{t}} \frac{\mathbb{E}[||D^t||_1]}{n \varepsilon}\right)$.
> If we choose adaptive $D^t$ for all t, we achieve a similar utility bound as Asi et al. (2021). Please see the table in our shared responses to all reviewers for detailed comparisons.
>
> **[Generalization]** As the main focus of our work is optimization, we do not study generalization bounds in this work. Moreover, the interplay between generalization and adaptive methods or differential privacy are open problems. Empirically, we observe that DP$^2$ improves over baselines in terms of both training and test performance (Figures 4 and 7 in the main text, and Figures 8 and 10-12 in the appendix).
>
> **[Staleness Parameter and the Temporal Gradient Similarity]** Yes, DP$^2$ needs to tune the staleness parameter, as we studied both theoretically (Section 4.1)  and empirically (Section 5.2---Effects of s). We do not need to tune the temporal gradient similarity in experiments; it is a property of the algorithms, which would be used as a metric only in analysis.
>
> **[Experiment Details and Reproducibility]** We report detailed experimental setup and additional results in Appendix C, including hyperparameters, additional results with DP$^2$ using AdaGrad, and additional baselines. Indeed, IMDB and StackOverflow are two representative datasets where adaptivity/preconditioning is crucial due to the sparsity of gradients. These datasets are used to illustrate the effectiveness of our method, as adaptive optimizers can achieve significantly higher utility than SGD for these datasets in non-private training, and we see that our method can retain those benefits in private training. To facilitate reproducibility, we have also submitted anonymized code as part of our response; please see the shared responses to all reviewers.

---

> > ### Comment · Reviewer_wuVW · 2022-11-22
> > **Response to the rebuttal**
> >
> > I apologize for the delay. I am still trying to understand the proof. Regardless of the outcome of the review process, I would suggest the authors to clear up the proof. Perhaps give more exposition of the proof idea. I would give more details here as I go on. I am getting confidence with the correctness of the proof. Based on that and the response of the authors, I am updating my score.
> >
> > Thanks for helping me understand Theorem 2. It is most helpful. I will look in to the codebase soon enough.
> >
> > For the public-assisted database, there is a paper by Kairouz et al. (COLT 2021) that is also relevant to the submission.

---

> > > ### Author Response · Authors · 2022-11-23
> > > **Proof and relevant paper**
> > >
> > > Thank you for replying and updating the score! We agree that the presentation of the proofs needs to be improved. We’ve already made revisions to this in the updated paper, and are happy to respond to your future questions/concerns about the proofs and the codebase.
> > >
> > > Thanks for pointing to the related paper of Kairouz et al. (COLT 2021) ((Nearly) Dimension Independent Private ERM with AdaGrad Rates via Publicly Estimated Subspaces), which we already discussed and compared with in the original submission (footnote on Page 9, Appendix C.5). We found that in this theoretically-focused work, the accuracy improves only marginally beyond random guessing; see Figure 12 and Appendix C.5 for detailed implementation and discussions. Implementation of this baseline method can also be found in our codebase (under trainers/dp_adagrad_grad_projection.py).

---

### Author Response · Authors · 2022-11-15
**Response to All Reviewers (2)**

**[Non-Convex Results]** As discussed by Reviewer vZ2E and wuVW, despite benefits in practice, our non-convex analysis (Theorem 3) does not directly highlight the benefits of adaptive methods, similar to other prior work studying stochastic non-convex adaptive optimization [e.g., Zaheer et al., 2018; Ward et al., 2018; De et al., 2018; Alacaoglu et al., 2020]. We consider rigorously characterizing the benefits of adaptivity for non-convex problems an open problem. Proving such benefits in the case of delayed preconditioners is an even more difficult problem, which we leave as an interesting direction of future work.

**[Reproducibility]** As instructed by the ICLR website, we planned to submit code by posting an official and confidential comment to reviewers and ACs. However, we were not able to do that this year before the reviews were released due to ICLR-specific OpenReview configurations. We have instead submitted our code during this rebuttal period to help reproducibility. Experimental details (including hyperparameters) can be found in Appendix C of our submission, as well as our code.

[Nemirosvki et al., 2009] Robust stochastic approximation approach to stochastic programming  \
[Bassily et al., 2014] Private empirical risk minimization: Efficient algorithms and tight error bounds \
[Asi et al., 2021] Private adaptive gradient methods for convex optimization \
[Li et al., 2022] Private adaptive optimization with side information \
[Zaheer et al., 2018] Adaptive methods for nonconvex optimization \
[Ward et al., 2018] AdaGrad stepsizes: Sharp convergence over nonconvex landscapes \
[De et al., 2018] Convergence guarantees for RMSProp and ADAM in non-convex optimization and an empirical comparison to Nesterov acceleration \
[Alacaoglu et al., 2020] A new regret analysis for Adam-type algorithms \
[Mukkamala & Hein, 2017] Variants of rmsprop and adagrad with logarithmic regret bounds

---

### Author Response · Authors · 2022-11-15
**Response to All Reviewers (1)**

We thank all reviewers for their time and helpful comments. We first address shared concerns and then respond to specific comments below.

**[Clarification of Theorem 2]** Reviewer wuVW and hof5 both ask for clarifications regarding Theorem 2. In the original submission, we presented the complete convergence bound in Theorem 2 in order to clearly expose all dependencies on the delay parameter $s$ (as $\alpha$, $\kappa$, and $D^t$ are all related to $s$). However, we agree that this makes the bound a bit difficult to parse, and it would be useful to provide a simpler result. Our rate can be simplified as:

$\min_t \mathbb{E}[F(w^t)] - F(w^*) \leq O\left(\frac{1}{\sqrt{T}}\right) \max_{t \in T_{\upsilon}} \mathbb{E}[||v^t+\varepsilon||\_1] + \frac{1}{T} \sum_{t} \frac{1}{\sqrt{t}} \mathbb{E}[||N^t||^2\_{D^t}]$,

where $t \in T_{\upsilon}$ corresponds to adaptive iterations when switching. We see from the bound that the added privacy noise would be reduced from $\frac{1}{T} \sum_{t} \frac{1}{\sqrt{t}} \mathbb{E}[||N^t||^2]$ to $\frac{1}{T} \sum_{t} \frac{1}{\sqrt{t}} \mathbb{E}[||N^t||^2\_{D^t}]$, if $D^t$ is sparse (i.e., $||D^t||\_1 \ll d$) in adaptive steps. Theorem 2 suggests some constant improvements compared with DP-SGD when we switch a constant number of times. We will update Theorem 2 in the revision. Thank you both for this suggestion!


**[Compare Theorem 2 with previous results]** Thank you for the comments. In terms of convergence rates, our $O\left(\frac{1}{\sqrt{T}}\right)$ rate is the same as previous results for SGD (or DP-SGD) in convex cases with diminishing step sizes [Nemirosvki et al., 2009, Bassily et al., 2014]. In terms of the utility, for sample size $n$, taking $\sigma^2 = O\left(\frac{b^2 T}{n^2\varepsilon^2}\right)$ and $T=O\left(n^2\varepsilon^2\right)$, the previous results on private adaptive stochastic optimization [Asi et al., 2021, Li et al., 2022] have rates that are roughly (ignoring $\log(1/\delta)$):

$\min_t \mathbb{E}[F(w^t)] - F(w^*)  \leq g(t) + O\left(\frac{tr(C)}{n \varepsilon}\right)$,

where $g(t)$ denotes the rate of standard adaptive optimization without privacy (e.g.,  $O\left(\frac{\mathbb{E}[||v^T+\varepsilon||_1]}{\sqrt{T}}\right)$  for RMSProp [e.g., Mukkamala & Hein, 2017]), and the second term is due to rescaled noise under a (fixed) diagonal projection matrix $C$.

Under the same $\sigma^2$ and $T$, our rate can be roughly written as

$\min_t \mathbb{E}[F(w^t)] - F(w^*) \leq O\left(\frac{1}{\sqrt{T}} \max_{t \in T_{\upsilon}} \mathbb{E}[||v^t+\varepsilon||_1]\right) + O\left(\frac{1}{\sqrt{T}} \sum_t \frac{1}{\sqrt{t}} \frac{\mathbb{E}[||D^t||_1]}{n \varepsilon}\right)$.

Compared with previous results, our first term (due to adaptive optimization) changes from the standard RMSProp rate to a slightly worse $O\left(\frac{1}{\sqrt{T}} \max_{t \in T_{\upsilon}} \mathbb{E}[||v^t+\varepsilon||\_1]\right)$, due to switching and the use of delayed preconditioners. Our second term (due to privacy noise) changes to $O\left(\frac{1}{\sqrt{T}} \sum_t \frac{1}{\sqrt{t}} \frac{\mathbb{E}[||D^t||\_1]}{n \varepsilon}\right)$, where $D^t$ is either an all-ones vector or $\sqrt{v^t} + \varepsilon$. If $D^t = \sqrt{v^t} + \varepsilon$ for all iterations (running adaptive methods all the time), the second term becomes $O(\max_{t \in [T]} \frac{\mathbb{E}[||D^t||_1]}{n \varepsilon}$, which is the same as prior works. Therefore, while our rates can be better than DP-SGD, we have slightly inferior constants relative to prior work in private adaptive optimization because we switch between DP-SGD and private RMSProp. **However, a major caveat here is that these prior results for private adaptive optimization are achieved when using algorithms that access public data.** We don’t rely on any public data assumptions in the DP$^2$ framework. We will add a discussion section about this in the next version. The comparisons are summarized in the table below.


[Note: $d$: model dimension; $n$: total number of samples; $\varepsilon$: privacy parameter; $D$: preconditioner; $C$: a diagonal matrix capturing gradient geometry]

| method | common assumptions | additional assumptions | bounds |
| ----------- | ----------- |----------- | ----------- |
| DP-SGD [Bassily et al. 2014] |convex | N/A | $\frac{\sqrt{d}}{n \varepsilon}$ |
|PAGAN [Asi et al., 2021] (AdaGrad) | convex | public data | AdaGrad rate + $\frac{tr(C)}{n \varepsilon}$ |
|AdaDPS [Li et al., 2022] (RMSProp) | convex | public data / side information | RMSProp rate + $\frac{tr(C)}{n \varepsilon}$ |
|DP$^2$ (ours) (RMSProp) | convex | N/A | Delayed RMSProp rate + $\frac{1}{\sqrt{T}} O\left(\sum_t \frac{1}{\sqrt{t}} \frac{\mathbb{E}[\|\|D^t\|\|_1]}{n \varepsilon}\right)$ |

---

### Decision · Program_Chairs · 2023-01-20

**Decision:**

Accept: poster

**Justification For Why Not Higher Score:**

The paper has novel insights and strong experiments, but the theoretical contribution for the non-convex setting is more limited.

**Justification For Why Not Lower Score:**

The contribution is relevant to the community and it may lead to further progress. The experimental evaluation is strong.

**Metareview: Summary, Strengths And Weaknesses:**

The paper proposes an algorithm for private adaptive optimization that achieves improved privacy/utility tradeoffs without using auxiliary data. The algorithm is based on two main insights: using averaging of the past gradients in order to reduce the effect of the DP noise, and applying the preconditioner before clipping.

The reviewers agree that the two main insights are novel and interesting, and the experimental evaluation is very strong. Overall, the paper makes a good contribution to the area of private optimization that is relevant to the community.

The reviewers agree that the theoretical guarantees for the non-convex setting are a potential weakness of the paper. The reviewers were also concerned about the clarity of the exposition. The authors revised the paper based on the reviewers' feedback and addressed some of the main concerns. Although the revision is satisfactory, the reviewers encourage the authors to further revise the paper to improve the technical clarity and to make the proof more accessible.

**Note From Pc:**

if the above contains the word "oral" or "spotlight" please see: "oral" presentation means -> notable-top-5% and "spotlight" means -> notable-top-25%. As stated in our emails, we are disassociating presentation type from AC recommendations

**Summary Of Ac-Reviewer Meeting:**

We discussed the revisions to the exposition that the authors made based on the reviewers' feedback. The consensus was that the revision was satisfactory, although the exposition could be further improved. We also discussed the theoretical analysis and experiments. The consensus was that the analysis for the non-convex setting is a potential weakness of the paper, but the analysis of the convex setting is a good contribution. The reviewers agreed that weaknesses on the theoretical front were counterbalanced by strong empirical performance.